# UniRTL: Unifying Code and Graph for Robust RTL Representation Learning

## Abstract

Developing effective representations for register transfer level (RTL) designs is crucial for accelerating the hardware design workflow. Existing approaches, however, typically rely on a single data modality, either the RTL code or its associated graph-based representation, limiting the expressiveness and generalization ability of the learned representations. Particularly, graph-related methods often adopt data flow or register-level sub-circuits, both of which capture only partial information and thus provide an incomplete view of the design. In contrast, the control data flow graph (CDFG) offers a more comprehensive structural representation that preserves complete information, while the code modality explicitly encodes semantic and functional information. We argue that integrating these complementary modalities is essential for a thorough understanding of RTL designs. To this end, we propose UniRTL, a multimodal pretraining framework that learns unified RTL representations by jointly leveraging code and CDFG. UniRTL achieves fine-grained alignment between code and graph through mutual masked modeling and employs a hierarchical training strategy that incorporates a pretrained graph-aware tokenizer and staged alignment of text (*i.e.*, functional summary) and code prior to graph integration. We evaluate UniRTL on two downstream tasks, performance prediction and code retrieval, under multiple settings. Experimental results show that UniRTL consistently outperforms prior methods, establishing it as a more robust and powerful foundation for advancing hardware design automation.

## 1 Introduction

Register transfer level (RTL) is a critical abstraction in the electronic design automation (EDA) workflow that describes the flow of data between registers and the logical operations performed on that data. As the front end of hardware design, deriving effective RTL representations can substantially accelerate the entire design process. For instance, developing informative RTL representations for performance prediction enables hardware designers to obtain instant feedback on key quality metrics such as area and delay, bypassing the need for time-consuming logic synthesis (Sengupta et al., 2022; Moravej et al., 2025; Liu et al., 2025c). Beyond performance prediction, effective RTL representations also facilitate tasks like code retrieval (Liu et al., 2025d), which allows for the efficient identification and reuse of relevant design modules. With the recent proliferation of large language models (LLMs) for RTL code generation (Pei et al., 2024; Zhao et al., 2025; Liu et al., 2025a;b), the development of powerful representations for retrieval has become even more important. These representations play a pivotal role in retrieval-augmented generation (RAG) (Lewis et al., 2020), thereby potentially enhancing the performance of RTL code generation systems.

Despite achieving promising performance, current approaches to RTL representation learning typically rely on a single data modality, either the RTL code or its associated graph-based representation, limiting the expressiveness and generalization ability of the learned representations. For example, in the context of performance prediction, VeriDistill (Moravej et al., 2025) derives representations by feeding RTL code into LLMs specifically fine-tuned for RTL code generation and aggregating token-level embeddings for prediction. On the other hand, StructRTL (Liu et al., 2025c) constructs representations using a structure-aware self-supervised learning framework applied to the control data flow graph (CDFG) of RTL designs. Similarly, for the code retrieval tasks, DeepRTL2 (Liu et al., 2025d) generates embeddings directly from RTL code using its backbone LLM. While the code modality explicitly encodes semantic and functional information, the graph modality captures

critical structural relationships that are often opaque from code. To achieve a more comprehensive understanding of RTL designs and obtain more robust and powerful representations, it is essential to develop methods that can effectively bridge these two modalities with complementary information.

In the software domain, GraphCodeBERT (Guo et al., 2021) enhances code understanding by pre-training representations of programming languages with data flow information. Despite its effectiveness, the model exhibits several notable limitations. First, there is a weak alignment between code and data flow established by the variable-alignment task, which merely locates variable nodes in the code without capturing their full semantic relationships. Second, the data flow representation itself is limited, as its nodes are restricted to variables, thereby overlooking other critical elements like operators and control flow, which are essential for tasks such as performance prediction and code retrieval. Finally, the model directly feeds variable-level data flow nodes into a Transformer (Vaswani et al., 2017) without employing a graph-aware tokenizer, which may hinder its ability to capture the nuanced and intricate structural relationships inherent in the graph. Recently, CircuitFusion (Fang et al., 2025) has been proposed for constructing multimodal fused representations of RTL by incorporating code, structural graphs, and functional summaries. In contrast to GraphCodeBERT, which adopts a unified Transformer architecture, CircuitFusion first derives unimodal representations using three independent encoders, and subsequently integrates them through a cross-attention mechanism. Nevertheless, its alignment strategy remains coarse-grained, where it relies on contrastive learning between text-code and text-graph pairs while neglecting fine-grained alignment between code and graph—two modalities that contain more detailed and richer information.

To bridge this gap, we propose UniRTL, a novel multimodal pretraining framework that learns unified RTL representations by leveraging complementary modalities of RTL. UniRTL addresses the limitations of prior work by achieving fine-grained cross-modal alignment through mutual masked modeling. Following GraphCodeBERT Guo et al. (2021), UniRTL employs a unified Transformer architecture to integrate different modalities, thereby eliminating the complexity of designing modality-specific encoders and enabling more seamless interaction across different modalities. Meanwhile, UniRTL adopts a hierarchical training strategy: a graph-aware tokenizer is first pretrained to enable the Transformer to better capture the nuanced structural dependencies in the graph, and alignment between text (*i.e.*, functional summary) and code is performed before incorporating the graph, which maximizes data utilization given the greater availability of text-code pairs compared to graph data. Moreover, instead of relying on data flow, UniRTL leverages CDFGs, which preserve complete information without loss and can be faithfully converted back to code.

We evaluate UniRTL on two downstream tasks, *i.e.*, performance prediction and code retrieval, each under multiple settings. For performance prediction, we examine post-synthesis area and delay estimation both with and without the incorporation of netlist information, consistent with the setting of StructRTL Liu et al. (2025c). For code retrieval, we consider scenarios where the query is either text or code, following the setup of DeepRTL2 Liu et al. (2025d). Across all tasks and settings, UniRTL consistently outperforms previous methods, demonstrating the effectiveness of our framework.

## 2 RELATED WORKS

**RTL Representation Learning.** Register transfer level (RTL) is a critical abstraction in the hardware design workflow, typically expressed in hardware description languages (HDLs) such as Verilog to specify data transfers between registers and the associated logical operations. Modern hardware design is inherently complex and involves multiple stages: natural language specifications are first manually translated into HDLs, which are then synthesized into circuit elements. Hardware designers often must wait for the time-consuming logic synthesis process to generate netlists and evaluate quality metrics, making iterative refinement slow and costly. To mitigate this bottleneck, prior research on RTL representation learning has primarily focused on performance prediction. For example, Sengupta et al. (2022) employ a graph attention network (GAT) Veličković et al. (2018) on constructed CDFGs for delay and power prediction, while StructRTL Liu et al. (2025c) introduces a structure-aware self-supervised learning framework on CDFGs for post-synthesis area and delay prediction. VeriDistill Moravej et al. (2025), in contrast, derives RTL representations using LLMs specifically fine-tuned for RTL code generation Pei et al. (2024); Cui et al. (2024); Zhao et al. (2025); Liu et al. (2025a;b). Beyond performance prediction, DeepRTL2 Liu et al. (2025d) explores the task of code retrieval, motivated by the high reusability of hardware designs. Specifically, it develops a

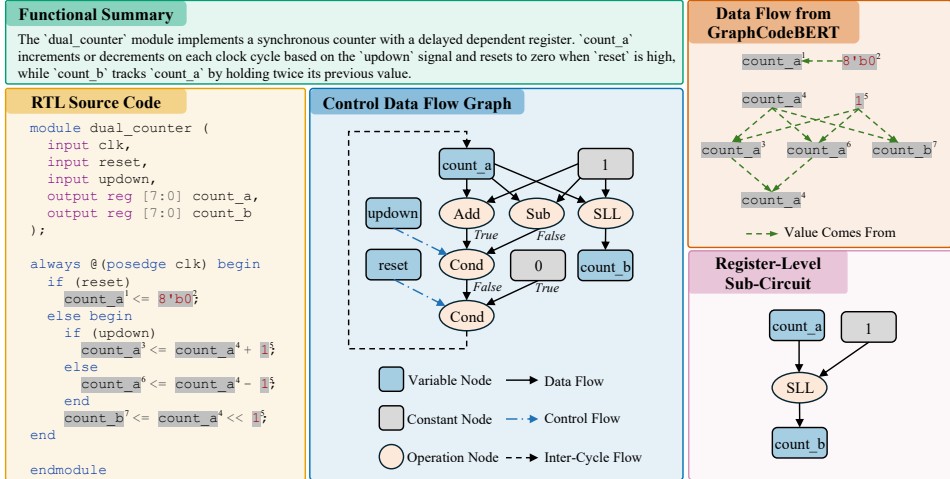

Figure 1: Example data point from our dataset, including RTL source code, and its corresponding functional summary and CDFG. For comparison, data flow (Guo et al., 2021) and register-level sub-circuit (Fang et al., 2025) are also shown, demonstrating the completeness of the constructed CDFG.

versatile model capable of both generation- and embedding-based tasks, where text and code embeddings are obtained from the backbone LLM. Despite these advances, existing approaches often rely on a single data modality, either the RTL code or its corresponding graph-based representation, which limits the expressiveness and generalization ability of the learned representations.

**Multimodal Representation Learning.** Multimodal representation learning aims to learn joint representations from multiple modalities, with recent advances spanning a variety of domains, including vision-language (Radford et al., 2021; Bao et al., 2022; Li et al., 2021; 2022; 2023; Jiang et al., 2025) and speech-text (Chuang et al., 2020; Tang et al., 2022; Yu et al., 2023). By integrating complementary information across modalities, these approaches enable the development of more robust and powerful representations for a wide range of tasks. Among existing works, the one most closely related to ours is GraphCodeBERT (Guo et al., 2021), which leverages data flow information to enhance code representation learning. However, its alignment strategy is limited: it merely identifies variable nodes in the code without capturing their full semantic relationships. Moreover, the employed data flow is incomplete, as it excludes critical elements such as operators and control flow, and the absence of a graph-aware tokenizer restricts the model's ability to capture the nuanced and intricate structural relationships inherent in the graph. Another relevant effort is CircuitFusion (Fang et al., 2025), which learns multimodal fused representations from RTL code, structural graphs, and functional summaries. Nevertheless, its alignment strategy relies on coarse-grained contrastive learning between text–code and text–graph pairs, while overlooking fine-grained alignment between code and graph. In addition, its dataset contains only 41 designs, and alignment is performed at the register sub-circuit level, which fails to capture the full semantics of entire modules or designs. In contrast, UniRTL achieves fine-grained alignment between code and graph through mutual masked modeling and is pretrained on a large-scale dataset. Furthermore, the adopted CDFGs preserve complete information without loss and can be faithfully converted back to code.

## 3 METHODOLOGY

In this section, we detail the dataset construction process, with particular emphasis on CDFG conversion, and present the overall dataset statistics. We then introduce the model architecture of UniRTL, highlighting both the mutual masked modeling alignment strategy and the hierarchical training strategy, in which a graph-aware tokenizer is first pretrained and text–code alignment is performed prior to incorporating the graph, thereby maximizing data utilization and enhancing model performance.

### 3.1 DATASET CONSTRUCTION

In this work, we collect datasets from multiple sources, including RTLCoder (Liu et al., 2024), MG-Verilog (Zhang et al., 2024), DeepRTL (Liu et al., 2025b), and DeepCircuitX (Li et al., 2025). These datasets contain original RTL designs paired with their corresponding functional summaries.

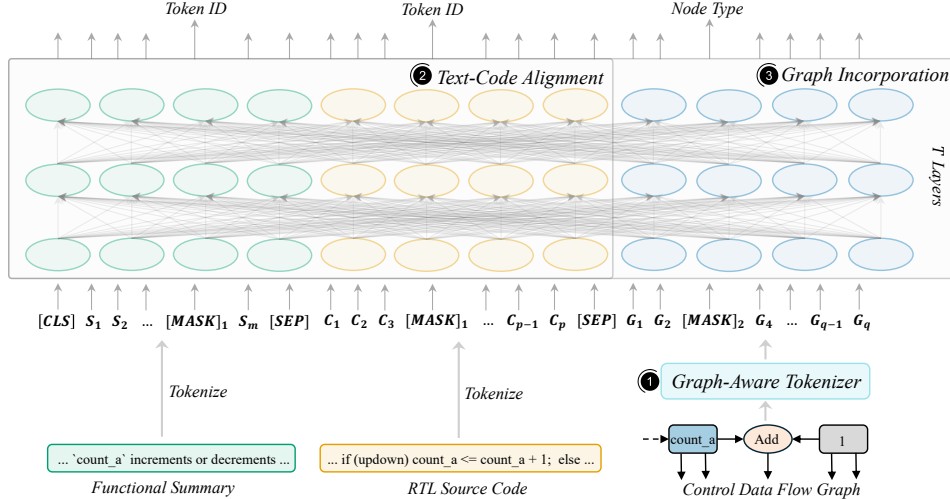

Figure 2: Overview of UniRTL. The framework achieves fine-grained cross-modal alignment via mutual masked modeling, and adopts a hierarchical training strategy, where a graph-aware tokenizer is first pretrained, and text-code alignment is performed prior to graph incorporation.

To construct CDFGs from RTL source code, we first compile the designs into RTL intermediate language (RTLIL) using Yosys (Wolf et al., 2013), a simplified form that preserves semantic completeness while reducing designs to basic assignment and register-transfer operations, thereby simplifying CDFG extraction. Next, we apply the Stagira Verilog parser Chen et al. (2023) to generate an abstract syntax tree (AST) from the RTLIL, and then traverse the AST to extract the CDFG. An example data sample is shown in Figure 1. Note that not all collected RTL designs can be successfully converted into CDFGs, as many originate from open-source GitHub repositories or are generated by LLMs and may contain syntax errors leading to compilation failures. Nevertheless, we retain these noisy samples for text–code alignment, enabling the model to learn more robust and generalizable representations while maximizing data utilization. In total, our dataset contains 132,008 RTL designs, of which 38,888 are successfully converted into CDFGs. Further analysis of the samples that fail to convert to CDFGs is provided in Appendix A.2.

## 3.2 MODEL ARCHITECTURE

We adopt a unified Transformer architecture as the backbone of UniRTL. Specifically, we use CodeBERT-base-mlm (Feng et al., 2020)[1] as our base model, pretrained on the CodeSearchNet (Husain et al., 2019) code corpus using masked language modeling (Devlin et al., 2019). This pretrained model provides UniRTL with rich prior knowledge of code. The overall framework of UniRTL is illustrated in Figure 2. UniRTL achieves fine-grained cross-modal alignment through mutual masked modeling, especially for the code and graph. Besides, to help the model better capture the nuanced and intricate structural relationships within the graph and maximize data utilization, we adopt a hierarchical training strategy, where a graph-aware tokenizer is first pretrained to encode structure-aware information in the CDFG, and text-code alignment is performed before the graph incorporation.

**Graph-Aware Tokenizer.** Unlike GraphCodeBERT (Guo et al., 2021), which directly feeds flattened variable nodes from the data flow into the Transformer, we design a graph-aware tokenizer tailored to encode structure-aware information from the CDFG. This enables the model to more effectively capture the nuanced and intricate structural relationships within the graph. The graph-aware tokenizer combines a graph isomorphism network (GIN) Xu et al. (2019) with a lightweight Transformer to jointly capture local structural dependencies and global contextual information. Specifically, given a graph $\mathcal{G} = \{\mathbb{V}, \mathbb{E}\}$, where $\mathbb{V}$ denotes the set of nodes and $\mathbb{E}$ the set of edges, we encode each node $v_i \in \mathbb{V}$ as:

$$\mathbf{H}_i = \text{concat}(\text{one-hot}(\text{type}(v_i)), \text{width}(v_i), \text{pca}(\phi_{\text{text}}(\text{desc}(v_i)))) \tag{1}$$

This representation concatenates the one-hot encoding of the node type, the node width, and the embedding of its textual description. $\phi_{\text{text}}$ denotes the text encoder, for which we use all-mpnet-base-

---

[1]https://huggingface.co/microsoft/codebert-base-mlm

v2[2]. To balance the contribution of different components, we apply principal component analysis (PCA) (Maćkiewicz & Ratajczak, 1993) to reduce the dimensionality of the description embedding from 768 to 32, matching the number of node types in our graphs. Incorporating description embeddings proves particularly effective, as it facilitates information alignment between the graph and code. After constructing the initial node embeddings, we feed the graph into a GIN to obtain node representations capturing local structural dependencies:

$$\mathbf{L}_i^{(k)} = \text{MLP}^{(k)} \left( \left(1 + \epsilon^{(k)}\right) \cdot \mathbf{L}_i^{(k-1)} + \sum_{j \in \mathcal{N}(i)} \mathbf{L}_j^{(k-1)} \right) \tag{2}$$

where $\mathbf{L}_i^{(0)} = \mathbf{H}_i$ is the initial embedding of node $v_i$, $\mathcal{N}(i)$ denotes the neighborhood of node $v_i$, and $\epsilon^{(k)}$ is a learnable scalar. After stacking $K$ GIN layers, we obtain the local structural embedding $\mathbf{L}_i = \mathbf{L}_i^{(K)}$. To incorporate global contextual information across the entire graph, we further process the GIN embeddings with a lightweight Transformer encoder, which takes $\{\mathbf{L}_i\}_{i \in \mathbb{V}}$ as input and produces refined node embeddings $\{\mathbf{G}_i\}_{i \in \mathbb{V}}$. The graph-aware tokenizer is pretrained with two objectives, structure-aware masked node modeling and edge prediction, enabling it to encode nuanced and intricate structural relationships within the graph. The embeddings $\{\mathbf{G}_i\}_{i \in \mathbb{V}}$ then serve as the input to UniRTL. For further details on the graph-aware tokenizer architecture and the pretraining tasks, please refer to Appendix A.3.

**Text-Code Alignment**. Since text-code pairs are more abundant than graph data, we first perform text-code alignment prior to incorporating the graph. This stage serves as a warm-up that familiarizes the model with RTL knowledge while maximizing data utilization. The alignment is achieved through mutual masked modeling. Specifically, the functional summary and RTL source code are tokenized into sequences $\{\mathbf{S}_i\}$ and $\{\mathbf{C}_i\}$, respectively. We then randomly mask 20% of the tokens, with 80% of the masked positions replaced by a special $[\text{MASK}]_1$ token, 10% by a random token, and 10% left unchanged. UniRTL is pretrained to recover these masked tokens by predicting their original token IDs. Since text and code encode complementary semantic information, masking one modality encourages the model to leverage the other for recovery, thereby promoting in-depth alignment between text and code.

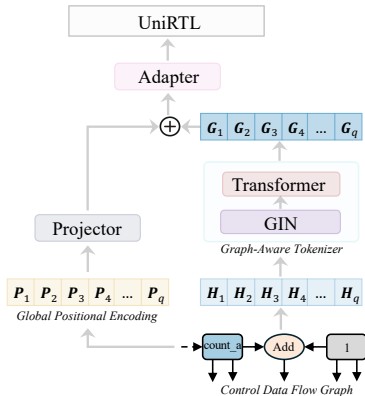

Figure 3: Preprocessing of the CDFG before being fed into UniRTL. Masking is applied to $\{\mathbf{G}_i\}_{i \in \mathbb{V}}$.

**Graph Incorporation.** After pretraining the graph-aware tokenizer and completing text-code alignment, we incorporate graph information into UniRTL to enable fine-grained alignment between code and graph. Specifically, given a graph, we first process it with the graph-aware tokenizer to obtain node embeddings $\{\mathbf{G}_i\}_{i \in \mathbb{V}}$ that capture the nuanced and intricate structural relationships within the graph. These embeddings are then fed into UniRTL, where alignment with text and code is achieved through mutual masked modeling. For text and code, we follow the same masking strategy used in text-code alignment. For the graph, 20% of the nodes are randomly selected and replaced with a learnable $[\text{MASK}]_2$ embedding. UniRTL is trained to recover the masked nodes by predicting their original node types, while simultaneously recovering masked text and code tokens. This joint objective encourages UniRTL to capture the full semantic relationships between code and graph. To preserve the graph's topological structure, we augment $\{\mathbf{G}_i\}_{i \in \mathbb{V}}$ with global positional encodings $\{\mathbf{P}_i\}_{i \in \mathbb{V}}$ (Rampášek et al., 2022) before feeding them into UniRTL. The global positional encodings are derived from the eigenvectors of the symmetric normalized graph Laplacian (Chung, 1997):

$$L = I - D_{\text{in}}^{-1/2} \left( \frac{A + A^{\text{T}}}{2} \right) D_{\text{out}}^{-1/2} \tag{3}$$

where $A$ is the adjacency matrix, and $D_{\text{in}}$ and $D_{\text{out}}$ denote the in-degree and out-degree matrices, respectively. The eigenvalues and eigenvectors of $L$ are then computed by solving:

$$L\mathbf{x} = \lambda\mathbf{x} \tag{4}$$

---

[2]https://huggingface.co/sentence-transformers/all-mpnet-base-v2

Table 1: Performance comparison of different methods on performance prediction tasks without the incorporation of netlist information. The best results are highlighted in bold.

| w/o Netlist Info | Area | | | | Delay | | | |
|---|---|---|---|---|---|---|---|---|
| | MAE↓ | MAPE↓ | $R^2$↑ | RRSE↓ | MAE↓ | MAPE↓ | $R^2$↑ | RRSE↓ |
| GAT | 0.5497 | 0.09 | 0.5857 | 0.6437 | 0.7327 | 0.13 | 0.6639 | 0.5797 |
| StructRTL | 0.3649 | **0.06** | 0.7463 | 0.5037 | 0.5414 | 0.10 | 0.7630 | 0.4868 |
| CodeV-DS-6.7B | 0.8967 | 0.17 | 0.4862 | 0.6973 | 0.6403 | 0.12 | 0.3905 | 0.7807 |
| CodeV-CL-7B | 0.7982 | 0.15 | 0.5755 | 0.6515 | 0.5620 | 0.10 | 0.5174 | 0.6947 |
| CodeV-QW-7B | 0.7229 | 0.13 | 0.6353 | 0.6039 | 0.5340 | 0.09 | 0.5277 | 0.6872 |
| DeepRTL2-Llama | 0.6988 | 0.12 | 0.6758 | 0.5694 | 0.5756 | 0.10 | 0.5017 | 0.7059 |
| DeepRTL2-DeepSeek | 0.7802 | 0.14 | 0.6225 | 0.6144 | 0.6357 | 0.11 | 0.4137 | 0.7657 |
| GraphCodeBERT | 0.8424 | 0.15 | 0.5207 | 0.6923 | 0.6109 | 0.11 | 0.3989 | 0.7753 |
| UniRTL | **0.3510** | **0.06** | **0.7682** | **0.4815** | **0.3384** | **0.06** | **0.7832** | **0.4656** |
| UniRTL (w/o code) | 0.3671 | 0.07 | 0.7546 | 0.4954 | 0.3584 | 0.06 | 0.7602 | 0.4897 |
| UniRTL (w/o graph) | 0.8818 | 0.15 | 0.5173 | 0.6948 | 0.6375 | 0.11 | 0.3839 | 0.7849 |

where $\{\lambda_i\}$ are the eigenvalues and $\{\mathbf{x}_i\}$ are the corresponding eigenvectors. We select the 16 smallest eigenvalues and their associated eigenvectors to construct the global positional encodings. Before integrating $\{\mathbf{P}_i\}_{i \in \mathbb{V}}$ with $\{\mathbf{G}_i\}_{i \in \mathbb{V}}$, a linear projection layer is applied to map the positional encodings to the same dimensionality as the node embeddings. Finally, an adapter is employed to project $\{\mathbf{G}_i\}_{i \in \mathbb{V}}$ into the joint text-code embedding space, thereby facilitating more effective cross-modal alignment. The overall process is illustrated in Figure 3.

## 4 EXPERIMENTAL RESULTS

In this section, we detail the experimental settings and present the results. We evaluate UniRTL on two representative downstream tasks, performance prediction and code retrieval, each under multiple settings. For performance prediction, we examine post-synthesis area and delay estimation, both with and without the incorporation of netlist information. For code retrieval, we consider scenarios where they query is either text or code. Across all tasks and settings, UniRTL consistently outperforms baseline methods, demonstrating the robustness and effectiveness of our framework.

### 4.1 BASELINE METHODS

For performance prediction, we consider several baselines: StructRTL (Liu et al., 2025c), VeriDistill (Moravej et al., 2025), and DeepRTL2 (Liu et al., 2025d). StructRTL derives RTL representations through a structure-aware self-supervised learning framework on CDFGs, while VeriDistill and DeepRTL2 obtain RTL representations by leveraging LLMs fine-tuned for RTL code generation to produce token-level embeddings, which are subsequently aggregated via mean or max pooling for prediction. Particularly, VeriDistill adopts the open-source Verilog LLM CodeV (Zhao et al., 2025), which offers three variants: CodeV-DS-6.7B, CodeV-CL-7B, and CodeV-QW-7B, fine-tuned from DeepSeek-Coder (Guo et al., 2024), CodeLlama Roziere et al. (2023), and CodeQwen (Bai et al., 2023), respectively. DeepRTL2 provides two variants, fine-tuned from Llama-3.1 (Grattafiori et al., 2024) and DeepSeek-Coder, respectively. We include all these variants in our comparison. In addition, we evaluate an end-to-end prediction method that employs a GAT directly over CDFGs for performance estimation (Sengupta et al., 2022). For code retrieval, we compare against state-of-the-art general-purpose text embedding models, including OpenAI's text-embedding-3-small and text-embedding-3-large (Neelakantan et al., 2022), NV-Embed-v2 (Lee et al., 2025) and GritLM-7B (Muennighoff et al., 2025), as well as customized RTL embedding models (DeepRTL2-Llama and DeepRTL2-DeepSeek). We also incorporate GraphCodeBERT (Guo et al., 2021) as a baseline for both tasks to highlight the necessity of our designs, including the use of complete graphs, the graph-aware tokenizer, and fine-grained alignment between code and graph. Importantly, we do not use it in its original, software-oriented form, but instead fine-tune it on the same RTL datasets and downstream tasks as UniRTL to ensure a fair comparison. We exclude CircuitFusion (Fang et al., 2025) from comparison due to the unavailability of released model checkpoints and insufficient details to enable faithful reproduction of their approach. A comparison of wall-clock training time for the different methods is provided in Appendix A.4.

Table 2: Performance comparison of different methods with the incorporation of netlist information. For reference, we also report the performance of the teacher model. The best results, excluding the teacher model, are highlighted in bold.

| w/ Netlist Info | Area | | | | Delay | | | |
|---|---|---|---|---|---|---|---|---|
| | MAE↓ | MAPE↓ | $R^2$↑ | RRSE↓ | MAE↓ | MAPE↓ | $R^2$↑ | RRSE↓ |
| PM Predictor | 0.2982 | 0.05 | 0.9334 | 0.2581 | 0.1688 | 0.03 | 0.9484 | 0.2272 |
| GAT | 0.4689 | 0.09 | 0.7954 | 0.4523 | 0.2926 | 0.05 | 0.8113 | 0.4344 |
| StructRTL | 0.3856 | 0.07 | 0.8676 | 0.3639 | 0.2381 | **0.04** | 0.8872 | 0.3359 |
| CodeV-DS-6.7B | 0.4896 | 0.09 | 0.7928 | 0.4552 | 0.3787 | 0.07 | 0.7235 | 0.5258 |
| CodeV-CL-7B | 0.4192 | 0.08 | 0.8225 | 0.4213 | 0.3208 | 0.06 | 0.7696 | 0.4800 |
| CodeV-QW-7B | 0.4397 | 0.08 | 0.8174 | 0.4273 | 0.3284 | 0.06 | 0.7687 | 0.4809 |
| DeepRTL2-Llama | 0.4540 | 0.08 | 0.8332 | 0.4085 | 0.3707 | 0.07 | 0.7445 | 0.5054 |
| DeepRTL2-DeepSeek | 0.4915 | 0.09 | 0.8287 | 0.4139 | 0.4014 | 0.07 | 0.7273 | 0.5222 |
| GraphCodeBERT | 0.6008 | 0.11 | 0.7578 | 0.4922 | 0.4289 | 0.07 | 0.6907 | 0.5561 |
| UniRTL | **0.3362** | **0.06** | **0.8879** | **0.3349** | **0.2302** | **0.04** | **0.8946** | **0.3247** |
| UniRTL (w/o code) | 0.3462 | **0.06** | 0.8741 | 0.3548 | 0.2764 | 0.05 | 0.8817 | 0.3439 |
| UniRTL (w/o graph) | 0.6121 | 0.11 | 0.7547 | 0.4953 | 0.4478 | 0.08 | 0.6775 | 0.5679 |

## 4.2 Experimental Setup

In this subsection, we detail the hyperparameter configurations for the model architecture and training process of UniRTL. UniRTL adopts the same architecture as its base model, CodeBERT-base-mlm (Feng et al., 2020), consisting of 12 Transformer layers with 12 attention heads per layer. During the text–code alignment stage, the base model is trained for 5 epochs on 4 NVIDIA L40 GPUs with a per-device batch size of 32. Training is performed using the AdamW optimizer (Loshchilov & Hutter, 2019) with a learning rate of 8e-5 and a weight decay of 0.01. To improve training stability, we employ a cosine learning rate scheduler with a warmup ratio of 0.03 and set the gradient accumulation steps to 8. After graph incorporation, the model is further trained for 300 epochs on 2 NVIDIA L40 GPUs with a per-device batch size of 16. All other hyperparameter settings remain the same as in the text–code alignment stage.

## 4.3 Performance Prediction

The experimental settings for performance prediction mainly follows StructRTL (Liu et al., 2025c). Specifically, we predict post-synthesis area and delay values, where RTL designs are synthesized and mapped to post-mapping netlists using Yosys (Wolf et al., 2013) and ABC (Brayton & Mishchenko, 2010) with the SkyWater 130nm technology library (Edwards, 2020). The area and delay values are then extracted from the generated netlists. For fine-tuning, we adopt the dataset from StructRTL, which consists of 13,200 designs split into training and validation sets with an 0.8:0.2 ratio. The task is formulated as a regression problem. After obtaining RTL representations with different methods, we fine-tune a three-layer multi-layer perceptron (MLP) to perform performance estimation. Additional details of the fine-tuning process are provided in Appendix A.5. For evaluation, we report four standard regression metrics: mean absolute error (MAE), mean absolute percentage error (MAPE), coefficient of determination ($R^2$), and root relative squared error (RRSE). Detailed definitions of these metrics are provided in Appendix A.6.

The performance prediction results of different methods are presented in Table 1. Notably, UniRTL consistently outperforms all baselines across all evaluation metrics for both post-synthesis area and delay prediction, establishing a new state of the art. Among the baselines, StructRTL achieves the strongest performance, highlighting the advantage of leveraging CDFGs over RTL source code, as CDFGs capture richer structural information that is critical for accurate performance estimation. In contrast, GraphCodeBERT, despite incorporating data flow information, performs significantly worse than other methods. This underperformance can be attributed to the limited scope of the data flow information it encodes, which is insufficient for this task, as well as its relatively small model size compared to LLM-based methods, resulting in weaker code embeddings. Importantly, UniRTL, with a model size comparable to GraphCodeBERT, surpasses not only GraphCodeBERT but also much larger LLM-based methods, underscoring the effectiveness and efficiency of our framework.

Table 3: Performance comparison of different methods on the natural language code search task, with F1 used as the main metric. The best scores are highlighted in bold.

| Model | Design Format | Precision↑ | Recall↑ | F1↑ (Main) |
|---|---|---|---|---|
| text-embedding-3-small | code | 0.254 | 0.350 | 0.277 |
| text-embedding-3-large | code | 0.350 | 0.442 | 0.375 |
| GritLM-7B | code | 0.393 | 0.475 | 0.414 |
| NV-Embed-v2 | code | 0.367 | 0.450 | 0.389 |
| DeepRTL2-Llama | code | 0.557 | 0.608 | 0.572 |
| DeepRTL2-DeepSeek | code | 0.532 | 0.592 | 0.547 |
| GraphCodeBERT | code & graph | 0.616 | 0.675 | 0.634 |
| UniRTL | code & graph | **0.650** | **0.692** | **0.662** |
| UniRTL (w/o graph) | code | 0.630 | 0.683 | 0.644 |

Additionally, we conduct an ablation study by removing the code and graph components of UniRTL, yielding two variants: UniRTL (w/o code) and UniRTL (w/o graph), respectively. We find that removing the graph component substantially degrades performance, underscoring the essential role of structural information encoded in CDFGs for performance prediction, while removing the code component results in a slight performance drop, indicating that code still provides complementary information that can enhance performance prediction.

To further enhance performance prediction, VeriDistill (Moravej et al., 2025) and StructRTL (Liu et al., 2025c) adopt a knowledge distillation strategy that transfers low-level insights from netlists into the performance predictor, *i.e.*, the three-layer MLP. Following StructRTL, we collect synthesized post-mapping (PM) netlists and train a GIN to directly predict performance metrics from these netlists. Since the area and delay values are directly extracted from the PM netlists, this PM predictor achieves high accuracy and serves as the teacher model. We then freeze the PM predictor and incorporate a knowledge distillation loss during the fine-tuning of the three-layer MLP, enabling it to integrate low-level information from the netlists. Experimental results with the incorporation of netlist information are reported in Table 2. As shown, incorporating netlist information improves the performance of all methods. Nevertheless, UniRTL achieves state-of-the-art performance by surpassing all baselines across all evaluation metrics for both area and delay prediction, further demonstrating the robustness of our framework. For additional details on the knowledge distillation process, please refer to Appendix A.7.

## 4.4 CODE RETRIEVAL

For code retrieval, we consider two scenarios in which the query is either text or code. Specifically, we adopt the settings of DeepRTL2 (Liu et al., 2025d), corresponding to its natural language code search and functionality equivalence checking tasks, respectively.

**Natural Language Code Search.** Natural language code search aims to retrieve relevant code snippets from a large codebase given natural language queries. We formulate it as a retrieval problem using the bitext mining setting of the MTEB benchmark (Muennighoff et al., 2022). Specifically, the input for this task consists of a tuple $(\mathcal{S}, \mathcal{R})$, where $\mathcal{S}$ denotes a list of functional summaries in natural language and $\mathcal{R}$ the corresponding RTL designs. In this work, elements of $\mathcal{R}$ may be provided either as RTL code alone or as "code & graph", where each RTL design includes both the code and its associated CDFG. During evaluation, all queries $\{\mathcal{S}_i\}$ and candidates $\{\mathcal{R}_i\}$ are embedded into fixed-length vectors. For each query $\mathcal{S}_i$, cosine similarity is computed against all candidates, and the index $j = \arg\max_k \cos(\mathcal{S}_i, \mathcal{R}_k)$ is selected. The retrieved $\mathcal{R}_j$ is regarded as the prediction for $\mathcal{S}_i$, while the corresponding $\mathcal{R}_i$ serves as the ground truth. For training and evaluation, we use the dataset and benchmark provided by DeepRTL2, with the modification that designs failing to convert successfully into CDFGs are removed to ensure fairness. We adopt three evaluation metrics: Precision, Recall, and F1, with F1 serving as the main metric. Further details of the experimental setup for this task are provided in Appendix A.8.

The experimental results are presented in Table 3. UniRTL consistently outperforms all baseline methods across all evaluation metrics, demonstrating the effectiveness of our framework. When restricted to the code-only format (UniRTL w/o graph), performance significantly degrades, highlighting the importance of incorporating graph information. Furthermore, UniRTL's improvements over

Table 4: Performance comparison of different methods on the functionality equivalence checking task, with average precision (AP) as the main metric. The best results are highlighted in bold.

| Model | Design Format | AP↑ (Main) | Accuracy↑ | F1↑ | Precision↑ | Recall↑ |
|---|---|---|---|---|---|---|
| text-embedding-3-small | code | 0.543 | 0.613 | 0.696 | 0.545 | 0.960 |
| text-embedding-3-large | code | 0.564 | 0.587 | 0.687 | 0.553 | 0.907 |
| GritLM-7B | code | 0.599 | 0.640 | 0.724 | 0.587 | 0.947 |
| NV-Embed-v2 | code | 0.554 | 0.607 | 0.667 | 0.547 | 0.853 |
| DeepRTL2-Llama | code | 0.646 | 0.695 | 0.737 | 0.597 | 0.964 |
| DeepRTL2-DeepSeek | code | 0.631 | 0.640 | 0.729 | 0.587 | 0.960 |
| GraphCodeBERT | code & graph | 0.730 | 0.733 | **0.753** | 0.613 | **0.973** |
| UniRTL | code & graph | **0.745** | **0.747** | **0.753** | **0.734** | 0.773 |
| UniRTL (w/o graph) | code | 0.712 | 0.667 | 0.717 | 0.577 | 0.947 |

GraphCodeBERT demonstrate the benefits of our fine-grained cross-modal alignment, hierarchical training strategy, and the integration of complete graph information. Interestingly, GraphCodeBERT even underperforms the variant of UniRTL where no graph is incorporated, which we hypothesize may be due to its targeted variable-alignment task interfering with the alignment between text and code, thereby hindering performance on natural language code search.

**Functionality Equivalence Checking.** Functionality equivalence checking aims to determine whether two different RTL implementations exhibit identical behavior despite structural differences. This task follows the pair classification setting of the MTEB benchmark. Specifically, the input for this task consists of $N$ pairs of RTL designs, $\{(\mathcal{R}_1^{(1)}, \mathcal{R}_1^{(2)})\}_{i=1}^N$, where each design can be represented either as code alone or as "code & graph". For each pair $(\mathcal{R}_1^{(1)}, \mathcal{R}_1^{(2)})$, the model is expected to determine whether they are functionally equivalent by calculating the cosine similarity between their embedding vectors. For training and evaluation, we adopt the dataset and benchmark provided by DeepRTL2, excluding designs that cannot be successfully converted to CDFGs to ensure fair evaluation. We report five evaluation metrics for this task: Average Precision (AP), Accuracy, F1, Precision, and Recall, with AP serving as the main metric. Further details on the experimental setup for this task are provided in Appendix A.9.

The performance comparison of different methods on the functionality equivalence checking task is presented in Table 4. UniRTL significantly outperforms all baseline methods on the main evaluation metric, further demonstrating the effectiveness and robustness of our framework. Removing the graph component (UniRTL w/o graph) leads to a substantial performance degradation, highlighting the importance of graph incorporation. Moreover, GraphCodeBERT performs better than the variant of UniRTL where no graph is incorporated, indicating that incorporating the data flow information can enhance the performance of functionality equivalence checking. However, UniRTL's superior performance over GraphCodeBERT demonstrates that merely leveraging data flow is insufficient; instead, dedicated strategies are essential to integrate the complete graph information, further validating the contributions of the various components in our framework. Additionally, we provide an ablation study of code–graph alignment strategies in Appendix A.11, and a qualitative analysis of code–graph relationships in Appendix A.12.

## 5 CONCLUSION

In this work, we introduce UniRTL, a multimodal pretraining framework that unifies RTL code and CDFGs for robust RTL representation learning. Unlike prior approaches that rely on simplified data flows or register-level sub-circuits, UniRTL leverages CDFGs that preserve complete design information and can be faithfully converted back to code. Furthermore, instead of establishing only weak code-graph alignment through contrastive objectives, UniRTL achieves fine-grained cross-modal alignment through mutual masked modeling. To better capture the nuanced and intricate structural dependencies within graphs, UniRTL employs a hierarchical training strategy: a graph-aware tokenizer is first pretrained, and text–code alignment is performed as a warm-up stage to maximize data utilization before incorporating the graph. We evaluate UniRTL on two representative downstream tasks, performance prediction and code retrieval, each under multiple settings. Experimental results demonstrate that UniRTL consistently outperforms existing baseline methods across all tasks and settings, validating its robustness and effectiveness. Overall, UniRTL establishes a more general and powerful foundation for advancing hardware design automation.

ETHICS STATEMENT

We have read the ICLR Code of Ethics[3] and are committed to adhering to it. Specifically, all source RTL designs are collected from open-source repositories under appropriate licenses, and dataset processing is conducted using open-source tools.

REPRODUCIBILITY STATEMENT

We have made efforts to ensure the reproducibility of this work. Specifically, Section 3.1 provides a detailed description of the dataset construction process, with particular emphasis on the CDFG generation. Section 4.2 further outlines the hyperparameters used in our experiments. In addition, we release the source code along with the training and evaluation datasets through an anonymous GitHub repository: `https://anonymous.4open.science/r/UniRTL-0EAE`.

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

# A APPENDIX

## A.1 THE USE OF LARGE LANGUAGE MODELS (LLMs)

We acknowledge the use of LLMs in the course of this work. Specifically, the LLM-powered programming tool Cursor[4] was employed to assist with implementation, while GPT-5[5] was used during manuscript preparation to correct grammar and refine phrasing.

## A.2 ANALYSIS OF CDFG CONVERSION FAILURES

In Section 3.1, we note that not all collected RTL designs can be successfully converted into CDFGs. Many designs originate from open-source GitHub repositories or are generated by LLMs and contain syntax errors that lead to compilation failures. In total, our dataset contains 132,008 RTL designs, of which 38,888 are successfully converted into CDFGs. Importantly, the 38,888 designs that successfully convert to CDFGs do not constitute a cherry-picked subset of simpler or cleaner RTL. In practice, most conversion failures arise from low-quality designs in existing open-source corpora, which would also fail standard EDA toolchains. Consequently, our filtering primarily enforces basic syntactic and compilation validity rather than favoring structurally simple circuits, and the resulting corpus still spans a broad range of design sizes, structural complexity, and coding styles. Nevertheless, we retain the noisy samples that fail CDFG conversion for text-code alignment, enabling the model to learn more robust and generalizable representations while maximizing data utilization. Empirical validation of this design choice is provided in Appendix A.10.

## A.3 DETAILS OF THE GRAPH-AWARE TOKENIZER

The graph-aware tokenizer integrates a graph isomorphism network (GIN) (Xu et al., 2019) with a lightweight Transformer encoder to jointly capture local structural dependencies and global contextual information. It is pretrained with two objectives, structure-aware masked node modeling and edge prediction, which enable it to capture the nuanced and intricate structural relationships within the graph. Specifically, given the initial node embeddings $\{\mathbf{H}_i\}_{i \in \mathbb{V}}$, the graph is first processed by the GIN to obtain $\{\mathbf{L}_i\}_{i \in \mathbb{V}}$ that encode local structural dependencies. These embeddings are then passed through the Transformer encoder to produce refined node embeddings $\{\mathbf{G}_i\}_{i \in \mathbb{V}}$. For structure-aware masked node modeling, we randomly replace 20% of nodes with a special learnable [MASK] embedding at the post-GIN level and use the Transformer encoder to recover masked nodes by predicting their original node types. Following StructRTL (Liu et al., 2025c), we adopt the class-balanced focal loss Cui et al. (2019) for this task to mitigate the node-type imbalance problem and denote the loss as $\mathcal{L}_{mnm}$. For edge prediction, the refined node embeddings $\{\mathbf{G}_i\}_{i \in \mathbb{V}}$ are used to predict the existence of edges between nodes. Since the Transformer encoder discards explicit connectivity, which can be viewed as if all edges are masked, we sample 20% of true edges as positive samples and an equal number of non-existing edges as negative samples in each iteration. The task is formulated as a binary classification problem, where we concatenate the final embeddings of the source and target nodes and use a three-layer multi-layer perceptron (MLP) to predict whether an edge exists between them. The cross-entropy loss is employed for this task, and the loss is denoted as $\mathcal{L}_{ep}$. Overall, the graph-aware tokenizer is pretrained with the loss:

$$\mathcal{L} = \gamma \cdot \mathcal{L}_{mnm} + (1 - \gamma) \cdot \mathcal{L}_{ep} \tag{5}$$

where $\gamma$ balances these two pretraining tasks, with $\gamma = 0.5$ in our experiments.

When node embeddings are flattened for input into the Transformer encoder, the graph's topological information is lost. To mitigate this issue, we incorporate global positional encodings into the post-GIN node embeddings $\{\mathbf{L}_i\}_{i \in \mathbb{V}}$ before feeding them into the Transformer encoder. The construction and application of these global positional encodings are described in Section 3.2.

The graph-aware tokenizer employs an 8-layer GIN and an 8-layer Transformer encoder, with 4 attention heads per Transformer layer. It is pretrained for 2,000 epochs with a batch size of 16 on a single NVIDIA L40 GPU, using AdamW (Loshchilov & Hutter, 2019) with a learning rate of 2e-5 and weight decay of 1e-4.

---

[4]https://cursor.com
[5]https://chatgpt.com

Table 5: Wall-clock pretraining and fine-tuning time and hardware for the baselines and UniRTL.

| Model | Hardware | Training time |
|---|---|---|
| VeriDistill | 8× NVIDIA V100 | 12 h |
| DeepRTL2 | 8× NVIDIA A800 | 70 h |
| GraphCodeBERT | 16× NVIDIA V100 | 83 h |
| GritLM-7B | 64× NVIDIA A100 | 48 h |
| text-embedding-3-small / -large | N/A | N/A |
| NV-Embed-v2 | N/A | N/A |
| GAT (graph-only baseline) | 1× NVIDIA L40 | 1 h |
| StructRTL | 1× NVIDIA L40 | 40 h |
| UniRTL | 2× NVIDIA L40 | 45 h |

Table 6: Hyperparameter configurations employed during fine-tuning for code retrieval tasks.

| Hyperparameter | Value |
|---|---|
| finetuning_type | full |
| temperature | 0.05 |
| normalize | true |
| optimizer | AdamW |
| learning_rate | 5e-5 |
| weight_decay | 0.01 |
| batch_size | 64 |
| epochs | 8 |
| lr_scheduler_type | cosine |
| warmup_ratio | 0.03 |
| gradient_accumulation_steps | 8 |

(a) Natural language code search

| Hyperparameter | Value |
|---|---|
| finetuning_type | full |
| temperature | 0.05 |
| normalize | true |
| optimizer | AdamW |
| learning_rate | 5e-5 |
| weight_decay | 0.01 |
| batch_size | 16 |
| epochs | 16 |
| lr_scheduler_type | cosine |
| warmup_ratio | 0.03 |
| gradient_accumulation_steps | 8 |
| max_hard_negatives | 3 |

(b) Functionality equivalence checking

After pretraining, the graph-aware tokenizer achieves evaluation accuracies of 85.04% on the structure-aware masked node modeling task and 99.68% on the edge prediction task. Following UniRTL pretraining, the masked node recovery accuracy further improves to 97.57%, demonstrating that incorporating code information enhances recovery performance and validates the effectiveness of our alignment strategy. We do not incorporate the edge prediction task during the pretraining of UniRTL since this task is relatively simple, converge quickly to high accuracy, and has negligible impact on the final model performance.

## A.4 TRAINING TIME COMPARISON

We report the wall-clock pretraining and fine-tuning time, together with the associated hardware, for UniRTL and the baselines in Table 5. Compared with LLM-based encoders such as GritLM-7B and DeepRTL2, UniRTL requires substantially less compute: it is trained on only 2× NVIDIA L40 GPUs for approximately 45 hours, whereas DeepRTL2 uses 8× NVIDIA A800 GPUs for 70 hours and GritLM-7B uses 64× NVIDIA A100 GPUs for 48 hours. GraphCodeBERT also incurs a higher training cost, requiring 16× NVIDIA V100 GPUs for 83 hours. At the same time, UniRTL is more computationally demanding than lightweight graph-only baselines such as GAT and StructRTL, reflecting the added capacity introduced by our joint code–graph modeling. However, UniRTL delivers substantially better performance than both GAT and StructRTL on the performance prediction tasks, indicating that the additional training cost is efficiently translated into meaningful quality gains. Overall, these results show that UniRTL attains strong performance with a moderate training budget that remains significantly lower than the compute required by large LLM-based encoders.

## A.5 FINE-TUNING FOR PERFORMANCE PREDICTION

After obtaining RTL representations from different methods, we fine-tune a three-layer MLP for performance prediction. Because the dimensionality of RTL representations varies across methods,

we first project them into a 512-dimensional space before feeding them into the MLP, which has a hidden layer size of 256. Given that area and delay values have large magnitudes and exhibit substantial variance across designs, we follow VeriDistill (Moravej et al., 2025) and StructRTL (Liu et al., 2025c) to apply a logarithm transformation to these values, making the target distribution more suitable for model learning. This transformation does not affect the practical utility of the predictor, as we are more concerned with the relative quality of different designs.

For training, we adopt the log-cosh loss (Saleh & Saleh, 2022), which is robust to outliers. The three-layer MLP predictors are trained for 600 epochs on a single NVIDIA L40 GPU with a batch size of 256, using the Adam optimizer (Kingma & Ba, 2015) with a learning rate of 1e-4 and a weight decay of 1e-5. Under this setup, all models are trained until full convergence.

## A.6 EVALUATION METRICS FOR PERFORMANCE PREDICTION

For performance prediction evaluation, we employ four standard regression metrics: mean absolute error (MAE), mean absolute percentage error (MAPE), coefficient of determination ($R^2$), and root relative squared error (RRSE). Given predicted values $\hat{y}_i$ and ground truth values $y_i$ for $i \in [1, N]$, these metrics are defined as:

$$\text{MAE} = \frac{1}{N} \sum_{i=1}^{N} |\hat{y}_i - y_i| \tag{6}$$

$$\text{MAPE} = \frac{1}{N} \sum_{i=1}^{N} \left| \frac{\hat{y}_i - y_i}{y_i} \right| \tag{7}$$

$$R^2 = 1 - \frac{\sum_{i=1}^{N} (\hat{y}_i - y_i)^2}{\sum_{i=1}^{N} (y_i - \bar{y})^2} \tag{8}$$

$$\text{RRSE} = \sqrt{\frac{\sum_{i=1}^{N} (\hat{y}_i - y_i)^2}{\sum_{i=1}^{N} (y_i - \bar{y})^2}} \tag{9}$$

where $\bar{y}$ denotes the mean of the ground truth values.

## A.7 PERFORMANCE PREDICTION WITH NETLIST INFORMATION

To further enhance performance prediction, we incorporate a knowledge distillation strategy that transfers low-level insights from post-mapping (PM) netlists into the RTL-stage performance predictors, *i.e.*, the three-layer MLP described in Appendix A.5. Following StructRTL (Liu et al., 2025c), we collect all synthesized PM netlists and train a GIN to directly predict performance metrics from these netlists. A PM netlist typically consists of interconnected logic cells defined in a technology library. To represent the PM netlist, we initialize each cell's embedding as the concatenation of its one-hot cell type encoding, logic truth table, and associated area and pin delay information. These embeddings are then processed by the GIN, followed by joint mean and max pooling to produce a graph-level representation, which is subsequently fed into a three-layer MLP for performance estimation. After training the PM predictor, we freeze its parameters and introduce a knowledge distillation loss during the training of the RTL-stage predictor, aligning the final-layer activations of the RTL-stage predictor ($z_{\text{RTL}}^{-1}$) with those of the PM predictor ($z_{\text{PM}}^{-1}$). The knowledge distillation loss is defined as:

$$\mathcal{L}_{kd} = \alpha \cdot \mathcal{L}_{cos}(z_{\text{RTL}}^{-1}, z_{\text{PM}}^{-1}) + (1 - \alpha) \cdot \mathcal{L}_{mse}(z\text{RTL}^{-1}, z_{\text{PM}}^{-1}), \tag{10}$$

where $\mathcal{L}_{cos}$ denotes the cosine similarity loss, $\mathcal{L}_{mse}$ the mean squared error (MSE) loss, and $\alpha$ balances the contribution of these two loss terms, set to 0.7 in our experiments.

The final loss for the RTL-stage predictor combines this distillation term with the log-cosh loss described in Appendix A.5:

$$\mathcal{L}_{pred} = \beta \cdot \mathcal{L}_{log\_cosh} + (1 - \beta) \cdot \mathcal{L}_{kd} \tag{11}$$

where $\beta$ is set to 0.5 in our experiments.

The adopted GIN consists of 20 layers with residual connections and is trained for 1,000 epochs using the log-cosh loss, a batch size of 16, and the same optimizer configuration as the RTL-stage predictor, on a single NVIDIA L40 GPU. It is important to note that the PM predictor is only used during training as a teacher; during inference, only the RTL-stage performance predictor is retained.

## A.8 EXPERIMENTAL SETUP FOR NATURAL LANGUAGE CODE SEARCH

We employ different strategies to obtain embeddings for functional summaries and RTL designs when evaluating different models on the natural language code search task. Specifically, for general-purpose text embedding models and customized RTL embedding models, we directly use their pre-trained weights and provided APIs to generate embeddings of the functional summaries and RTL designs. Since GritLM-7B (Muennighoff et al., 2025) and NV-Embed-v2 (Lee et al., 2025) are trained under an instruction-tuning paradigm, we prepend the instruction "*Given a high-level functional summary, retrieve the corresponding RTL code.*" in the model-specific template when extracting embeddings of functional summaries.

For GraphCodeBERT (Guo et al., 2021) and UniRTL, we take the last hidden state of the first token, *i.e.*, the `[CLS]` token, as the embedding vector for both $\mathcal{S}_i$ and $\mathcal{R}_i$ (in either code-only or "code & graph" format). These models are fine-tuned on this task using contrastive learning prior to evaluation. For all model variants, we keep the dataset and hyperparameter settings consistent during fine-tuning to ensure a fair comparison. We adopt the InfoNCE loss (Oord et al., 2018) for downstream fine-tuning on this task:

$$\mathcal{L}_{\text{nlcs}} = -\frac{1}{M} \sum_{i=1}^{M} \log \left( \frac{\exp\left(\frac{\cos(f_\theta(\mathcal{S}_i), f_\theta(\mathcal{R}_i))}{\tau}\right)}{\sum_{j=1}^{M} \exp\left(\frac{\cos(f_\theta(\mathcal{S}_i), f_\theta(\mathcal{R}_j))}{\tau}\right)} \right) \tag{12}$$

where $M$ is the batch size, $\mathcal{S}_i$ is the $i$-th functional summary in the batch, $\mathcal{R}i$ is the corresponding RTL design, $f_\theta$ is the embedding function, and $\tau$ is the temperature hyperparameter.

Let the evaluation benchmark be $(\mathcal{S}, \mathcal{R})$, where both $\mathcal{S}$ and $\mathcal{R}$ contain $N$ samples. During evaluation, the task is formulated as an $N$-class classification problem. Each $\mathcal{S}_i$ is treated as a sample belonging to class $i$, and the embedding model $f_\theta$ predicts its class as $\arg\max_k \cos(f_\theta(\mathcal{S}_i), f_\theta(\mathcal{R}_k))$.

Evaluation metrics for this task include Precision, Recall and F1, following the standard paradigm of multi-class classification, with F1 serving as the main metric. Downstream fine-tuning for this task is conducted on a single NVIDIA L40 GPU, and the hyperparameter settings are provided in Table 6a. An illustrative data example for this task is shown in Listing 1, comprising a high-level functional summary of an arithmetic logic unit (ALU) and its corresponding Verilog implementation.

## A.9 EXPERIMENTAL SETUP FOR FUNCTIONALITY EQUIVALENCE CHECKING

For the functionality equivalence checking task, we follow a strategy similar to that used for natural language code search (see Appendix A.8) to obtain embeddings of RTL designs. General-purpose text embedding models and customized RTL embedding models are evaluated without tuning their original parameters. For instruction-tuned text embedding models GritLM-7B (Muennighoff et al., 2025) and NV-Embed-v2 (Lee et al., 2025), we prepend the instruction "*Determine whether the given pair of RTL code snippets is functionally equivalent.*" to their model-specific templates to adapt their embeddings to this task.

For GraphCodeBERT (Guo et al., 2021) and UniRTL, we take the last hidden state of the `[CLS]` token as the embedding vector for each RTL design. These models are fine-tuned on this task using contrastive learning, where functionally inequivalent designs are used as hard negatives. To ensure fair comparison, all variants are fine-tuned under identical dataset and hyperparameter settings.

The fine-tuning dataset for this task is formatted as $\{(\mathcal{R}_i, \mathcal{E}_i, \mathcal{U}_i)\}_{i=1}^{N}$, where $\mathcal{R}_i$ is an RTL design, $\mathcal{E}_i$ is a corresponding RTL design with the same functionality, and $\mathcal{U}_i$ is a list of functionally inequivalent designs that serve as hard negatives. We adopt the InfoNCE loss Oord et al. (2018) with

Table 7: Performance comparison of UniRTL (w/o graph) and CodeBERT-based baselines on performance prediction tasks without incorporating netlist information. CodeBERT-IP denotes the CodeBERT (Incomplete Pretrain).

| Method | Area | | | | Delay | | | |
|---|---|---|---|---|---|---|---|---|
| | MAE↓ | MAPE↓ | $R^2$↑ | RRSE↓ | MAE↓ | MAPE↓ | $R^2$↑ | RRSE↓ |
| CodeBERT | 1.0011 | 0.17 | 0.3858 | 0.7837 | 0.7034 | 0.12 | 0.2539 | 0.8638 |
| CodeBERT-IP | 0.9514 | 0.16 | 0.4646 | 0.7317 | 0.6576 | 0.11 | 0.3650 | 0.7969 |
| UniRTL (w/o graph) | 0.8818 | 0.15 | 0.5173 | 0.6948 | 0.6375 | 0.11 | 0.3839 | 0.7849 |

Table 8: Performance comparison of UniRTL (w/o graph) and CodeBERT-based baselines on the natural language code search task, with F1 used as the main metric. CodeBERT-IP denotes the CodeBERT (Incomplete Pretrain).

| Model | Design Format | Precision↑ | Recall↑ | F1↑ (Main) |
|---|---|---|---|---|
| CodeBERT | code | 0.565 | 0.625 | 0.585 |
| CodeBERT-IP | code | 0.600 | 0.658 | 0.618 |
| UniRTL (w/o graph) | code | 0.630 | 0.683 | 0.644 |

hard negatives for downstream fine-tuning on this task:

$$\mathcal{L}_{\text{fec}} = -\frac{1}{M} \sum_{i=1}^{M} \log \left( \frac{\exp\left(\frac{\cos(f_\theta(\mathcal{R}_i), f_\theta(\mathcal{E}_i))}{\tau}\right)}{\sum_{j=1}^{M} \exp\left(\frac{\cos(f_\theta(\mathcal{R}_i), f_\theta(\mathcal{E}_j)))}{\tau}\right) + \sum_{j=1}^{M} \sum_{k=1}^{h_j} \left(\frac{\cos(f_\theta(\mathcal{R}_i), f_\theta(\mathcal{U}_j[k]))}{\tau}\right)} \right) \tag{13}$$

where $M$ is the batch size, $f_\theta$ is the embedding function, $\tau$ is the temperature hyperparameter, and $h_j = \min(\text{length}(\mathcal{U}_j), \text{max\_hard\_negatives})$, is the number of hard negatives used for sample $j$, controlled by the hyperparameter max_hard_negatives.

We evaluate models using five metrics: Average Precision (AP), Accuracy, F1, Precision, and Recall, with AP serving as the main metric. All evaluation metrics take as input a list of cosine similarity scores and binary labels, where 1 indicates functional equivalence and 0 indicates inequivalence. The threshold for functional equivalence is determined differently depending on the specific evaluation metric. The main metric, AP, requires no thresholding and is computed using the `average_precision_score` function in the Python `scikit-learn` library[6]. For accuracy, the threshold that maximizes classification accuracy is selected by enumerating over all possible thresholds. Specifically, we rank the similarity scores from highest to lowest, compute the accuracy at each possible threshold, and select the threshold that achieves the maximum accuracy. For F1, we similarly enumerate thresholds to identify the one that maximizes F1, and then report the corresponding F1, Precision, and Recall. This process ensures that we use the most appropriate threshold for each metric, allowing for accurate evaluation of the functionality equivalence. Our evaluation pipeline follows the pair-classification paradigm of the MTEB benchmark (Muennighoff et al., 2022), with implementation details available in the official MTEB GitHub repository[7]. Downstream fine-tuning for this task is performed on a single NVIDIA L40 GPU, and the hyperparameter settings are listed in Table 6b. An example training instance from the functionality equivalence checking dataset is shown in Listing 2. In this example, all three RTL designs share the same module name and interface (inputs and outputs), but differ in their internal implementations. The "Code" design serves as the query, the "Equal" design has equivalent functionality, and the "Unequal" design is not functionally equivalent to the query design despite structural similarity.

## A.10 LEVERAGING NOISY SAMPLES FOR TEXT-CODE ALIGNMENT

In Appendix A.2, we state that RTL samples that cannot be successfully converted into CDFGs are still used as noisy data for text-code alignment, enabling the model to learn more robust and

---

[6] https://scikit-learn.org/stable/modules/generated/sklearn.metrics. average_precision_score.html

[7] https://github.com/embeddings-benchmark/mteb

Table 9: Performance comparison of UniRTL (w/o graph) and CodeBERT-based baselines on the functionality equivalence checking task, with average precision (AP) as the main metric. CodeBERT-IP denotes the CodeBERT (Incomplete Pretrain).

| Model | Design Format | AP↑ (Main) | Accuracy↑ | F1↑ | Precision↑ | Recall↑ |
|---|---|---|---|---|---|---|
| CodeBERT | code | 0.617 | 0.633 | 0.712 | 0.586 | 0.907 |
| CodeBERT-IP | code | 0.668 | 0.633 | 0.682 | 0.521 | 0.987 |
| UniRTL (w/o graph) | code | 0.712 | 0.667 | 0.717 | 0.577 | 0.947 |

Table 10: Performance comparison of UniRTL variants with different code–graph alignment strategies on performance prediction tasks without incorporating netlist information. UniRTL (direct-combine) denotes a variant that simply concatenates the representations from the text–code-aligned encoder and the graph-aware tokenizer without explicit multimodal alignment pretraining.

| Method | Area | | | | Delay | | | |
|---|---|---|---|---|---|---|---|---|
| | MAE↓ | MAPE↓ | $R^2$↑ | RRSE↓ | MAE↓ | MAPE↓ | $R^2$↑ | RRSE↓ |
| UniRTL (w/o graph) | 0.8818 | 0.15 | 0.5173 | 0.6948 | 0.6375 | 0.11 | 0.3839 | 0.7849 |
| UniRTL (direct-combine) | 0.3629 | 0.06 | 0.7602 | 0.4897 | 0.3529 | 0.06 | 0.7624 | 0.4874 |
| UniRTL | 0.3510 | 0.06 | 0.7682 | 0.4815 | 0.3384 | 0.06 | 0.7832 | 0.4656 |

generalizable representations while maximizing data utilization. To assess whether this strategy truly improves learning and makes effective use of the available data, we compare UniRTL (w/o graph) with two CodeBERT-based baselines: the original CodeBERT-base-mlm model (CodeBERT) and CodeBERT-IP (Incomplete Pretrain), a variant obtained by pretraining and fine-tuning CodeBERT-base-mlm only on the 38,888 designs with valid CDFGs, while keeping all other settings fixed. As shown in Tables 7, 8 and 9, CodeBERT-IP consistently outperforms the vanilla CodeBERT baseline, demonstrating the effectiveness of our curated RTL dataset constructed from designs with valid CDFGs. Moreover, UniRTL (w/o graph), which further leverages noisy samples without CDFGs, surpasses both CodeBERT and CodeBERT-IP. This indicates that our hierarchical training strategy genuinely maximizes data utilization and yields better representations, rather than merely masking a data-quality problem.

## A.11 ABLATION STUDY OF CODE–GRAPH ALIGNMENT STRATEGIES

In addition to the modality ablations reported in the main paper, we further analyze how different code–graph alignment strategies affect UniRTL's performance. First, we note that GraphCodeBERT is already included as a strong baseline that reflects an alternative alignment mechanism between code and graph. To ensure a fair comparison, we fine-tune GraphCodeBERT on our dataset rather than using its original, software-oriented form. GraphCodeBERT aligns code and graph via a contrastive variable-alignment objective that encourages correspondence between variable nodes in the graph and code tokens. While effective, this objective primarily targets variable-level matching and does not explicitly model richer semantic relations between arbitrary tokens and CDFG nodes. By contrast, UniRTL adopts a mutual masked modeling objective across the code and CDFG modalities. Rather than aligning only variable mentions, UniRTL jointly reconstructs masked elements in one modality using information from the other, thereby encouraging finer-grained cross-modal alignment between code tokens and graph nodes. This design is intended to capture broader semantic and structural relationships beyond simple variable correspondences.

To more directly probe the effect of this mutual masked modeling objective, we conduct an additional ablation with a "direct-combine" baseline. In this variant, we simply concatenate the representations from the text–code-aligned encoder and the graph-aware tokenizer, without any explicit multimodal alignment pretraining between code and graph. All models are trained and evaluated under identical settings. As shown in Tables 10, 11, and 12, across all tasks, the direct-combine baseline improves over the corresponding unimodal variant, confirming that the mere presence of both modalities is beneficial. However, UniRTL with mutual masked modeling consistently achieves the best performance, providing clear and non-trivial gains over direct combine on natural language code search, functionality equivalence checking, and both area and delay prediction. These im-

Table 11: Performance comparison of UniRTL variants with different code–graph alignment strategies on the natural language code search task, with F1 used as the main metric. UniRTL (direct-combine) denotes a variant that simply concatenates the representations from the text–code-aligned encoder and the graph-aware tokenizer without explicit multimodal alignment pretraining.

| Model | Design Format | Precision↑ | Recall↑ | F1↑ (Main) |
|---|---|---|---|---|
| UniRTL (w/o graph) | code | 0.630 | 0.683 | 0.644 |
| UniRTL (direct-combine) | code & graph | 0.636 | 0.683 | 0.650 |
| UniRTL | code & graph | 0.650 | 0.692 | 0.662 |

Table 12: Performance comparison of UniRTL variants with different code–graph alignment strategies on the functionality equivalence checking task, with average precision (AP) as the main metric. UniRTL (direct-combine) denotes a variant that concatenates the representations from the text–code-aligned encoder and the graph-aware tokenizer without multimodal alignment pretraining.

| Model | Design Format | AP↑ (Main) | Accuracy↑ | F1↑ | Precision↑ | Recall↑ |
|---|---|---|---|---|---|---|
| UniRTL (w/o graph) | code | 0.712 | 0.667 | 0.717 | 0.577 | 0.947 |
| UniRTL (direct-combine) | code & graph | 0.737 | 0.687 | 0.723 | 0.595 | 0.920 |
| UniRTL | code & graph | 0.745 | 0.747 | 0.753 | 0.734 | 0.773 |

provements indicate that our alignment strategy contributes beyond mere multi-modality by enabling finer-grained cross-modal understanding between code and CDFG.

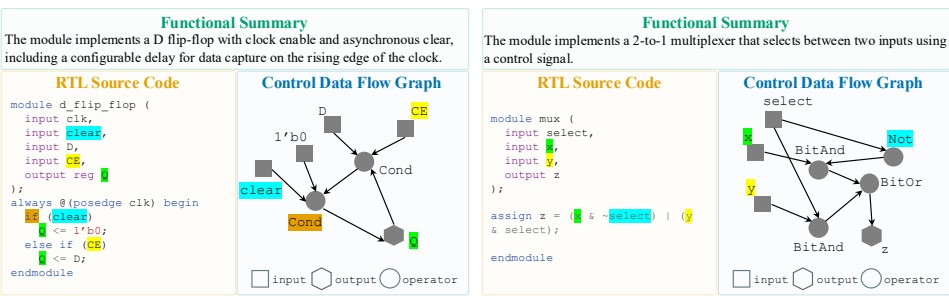

(a) Code-to-Graph Attention        (b) Graph-to-Code Attention

Figure 4: Qualitative analysis of code–graph relationships learned by mutual masked modeling. (a) Nodes with the highest attention scores from code tokens. (b) Code tokens with the highest attention scores from graph nodes. Corresponding code token–graph node pairs are highlighted with the same background color.

## A.12   QUALITATIVE ANALYSIS OF CODE–GRAPH RELATIONSHIPS

During the graph incorporation stage, the mutual masked prediction task requires the model to recover masked code tokens from the functional summary, surrounding code tokens, and graph nodes, and symmetrically to infer masked graph node types from the functional summary, neighboring graph nodes, and code tokens. To illustrate the nature of the resulting cross-modality alignment, we conduct a qualitative analysis of the code–graph relationships learned by the model. Specifically, we analyze cross-modal attention patterns in the pretrained UniRTL model. For each input triplet (functional summary, RTL source code, CDFG), we extract final-layer attention scores by summing the attention weights across all heads in the last Transformer layer. We then examine attention in a bilateral manner: (1) code-to-graph, where for each code token we identify the graph nodes receiving the highest attention from that token; and (2) graph-to-code, where for each graph node we identify the code tokens receiving the highest attention from that node. Figure 4 shows representative examples. In Figure 4(a), the code tokens corresponding to clear, CE, and Q attend strongly to their respective nodes in the CDFG. In Figure 4(b), the graph nodes x and y direct most of their attention to the corresponding code tokens x and y. These visualizations indicate that UniRTL learns to focus on semantically corresponding elements across modalities, thereby capturing meaningful alignments between specific code expressions and their associated CDFG nodes.

Listing 1: Functional summary of an ALU and its corresponding Verilog implementation.

```
Functional Summary:
The code defines an ALU that executes operations like add, subtract,
bitwise AND/OR, left/right shift, and bitwise NOT based on a control
signal. It processes two 8-bit inputs and produces an 8-bit result, with
a flag to indicate a zero output.
```

```verilog
Code:
module Alu(
    Alu_in1,
    Alu_in2,
    Alu_sel,
    Alu_zero_flg,
    Alu_out
);
    parameter wrd_size = 8,
              sel_width= 3;
    input [wrd_size-1:0] Alu_in1, Alu_in2;
    input [sel_width-1:0] Alu_sel;
    output reg [wrd_size-1:0] Alu_out;
    output Alu_zero_flg;
    localparam NOP = 3'b000,
               ADD = 3'b001,
               SUB = 3'b010,
               AND = 3'b011,
               OR  = 3'b100,
               SLT = 3'b101,
               SRT = 3'b110,
               NOT = 3'b111;
    assign Alu_zero_flg = ~|Alu_out;
    always @(*) begin
        case(Alu_sel)
            NOP:  Alu_out = 0;
            AND:  Alu_out = Alu_in1 & Alu_in2;
            OR:   Alu_out = Alu_in1 | Alu_in2;
            ADD:  Alu_out = Alu_in1 + Alu_in2;
            SUB:  Alu_out = Alu_in1 - Alu_in2;
            NOT:  Alu_out = ~Alu_in1;
            SLT:  Alu_out = Alu_in1 << Alu_in2;
            SRT:  Alu_out = Alu_in1 >> Alu_in2;
            default: Alu_out = 0;
        endcase
    end
endmodule
```

Listing 2: Example training sample for the functionality equivalence checking task.

```
Code:
module AND2_X4 (A1, A2, ZN);
    input A1;
    input A2;
    output ZN;
    and(ZN, A1, A2);
    specify
        (A1 => ZN) = (0.1, 0.1);
        (A2 => ZN) = (0.1, 0.1);
    endspecify
endmodule
```

```
Equal:
module AND2_X4 (A1, A2, ZN);
    input A1;
    input A2;
    output ZN;
    assign ZN = A1 & A2;
    specify
        (A1 => ZN) = (0.1, 0.1);
        (A2 => ZN) = (0.1, 0.1);
    endspecify
endmodule
```

```
Unequal:
module AND2_X4 (A1, A2, ZN);
    input A1;
    input A2;
    output ZN;
    wire nA1;
    wire nA2;
    wire nZN;
    nand(nZN, nA1, nA2);
    not(nA1, A1);
    not(nA2, A2);
    not(ZN, nZN);
    specify
        (A1 => ZN) = (0.1, 0.1);
        (A2 => ZN) = (0.1, 0.1);
    endspecify
endmodule
```

