# OpenReview forum: "UniRTL: Unifying Code and Graph for Robust RTL Representation Learning"
_ICLR.cc/2026/Conference — Submitted to ICLR 2026_

### Official Review · Reviewer_GRuV · 2025-10-26

**Soundness:** 3
**Presentation:** 2
**Contribution:** 2
**Rating:** 4
**Confidence:** 4

**Summary:**

This paper presents UniRTL, a multimodal pretraining framework designed to learn unified and robust representations for Register Transfer Level (RTL) hardware designs. The authors argue that existing methods are limited because they typically rely on only a single modality – either the RTL source code or a graph representation (like data flow graphs or register-level sub-circuits). These single modalities capture incomplete information; code provides semantic/functional details, while graphs offer structural insights. Furthermore, existing graph representations like data flow are often incomplete.

UniRTL aims to overcome these limitations by jointly leveraging RTL source code and its corresponding Control Data Flow Graph (CDFG), which the authors argue is a more comprehensive structural representation. The key components of UniRTL are: Unified Transformer Architecture, Graph-Aware Tokenizer, Mutual Masked Modeling, Hierarchical Training Strategy.

**Strengths:**

Strong Motivation for Multimodality: The paper makes a compelling case for why combining code and a comprehensive graph representation (CDFG) is necessary for robust RTL understanding. It clearly articulates the complementary nature of these modalities and the limitations of prior work using incomplete graphs (data flow) or only code.

Use of CDFG: Choosing CDFG over simpler graph forms like data flow or register sub-circuits is a significant strength. As argued (and illustrated in Figure 1), CDFG preserves more complete structural and control flow information, which is likely crucial for tasks like performance prediction.

**Weaknesses:**

Complexity: The overall framework is quite complex, involving multiple pretraining stages, a specialized graph tokenizer, and careful integration of three modalities (text summary, code, graph). This might raise concerns about the practicality of training and deploying such a model compared to simpler unimodal approaches.

CDFG Generation Failures: The paper notes that CDFG generation failed for a large portion of the collected RTL designs (only ~39k out of ~132k succeeded). While the failed samples were still used for text-code alignment, this highlights a potential practical limitation: the approach relies heavily on the availability and successful generation of high-quality CDFGs. The reasons for failure (syntax errors, tool limitations?) and their impact are not fully explored.

Limited Ablation on Alignment: While ablation studies show the benefit of including both code and graph (Table 1), there isn't a direct comparison between the mutual masked modeling alignment strategy and other potential alignment methods (e.g., contrastive loss between graph nodes and code tokens). This makes it slightly harder to isolate the specific contribution of the masking approach versus just having both modalities present.

Dependence on Base Code Model: The performance relies significantly on the underlying CodeBERT base model. It's unclear how much of the gain comes from the multimodal fusion itself versus simply starting with a strong code foundation model.

**Questions:**

CDFG Robustness: Could the authors elaborate on the CDFG generation failures? Were these failures concentrated in specific types of RTL designs (e.g., those generated by LLMs, or using specific Verilog constructs)? How robust is the CDFG generation process to variations in coding style or complexity? Does the model's performance correlate with the quality or completeness of the generated CDFG?

Complexity vs. Performance Trade-off: Given the complexity of the multi-stage training and the specialized graph tokenizer, how does UniRTL compare to simpler baselines (e.g., just fine-tuning a large code LLM like CodeLlama/DeepSeek-Coder directly on the downstream tasks) in terms of training efficiency and inference latency versus the observed performance gains?

Alternative Graph Representations: The paper argues strongly for CDFG. Have the authors experimented with or considered other graph representations that might be easier to generate reliably (even if less complete), such as Abstract Syntax Trees (ASTs) augmented with some data flow edges? How would UniRTL perform with these alternatives?

Nature of Code-Graph Alignment: The mutual masking forces the model to predict masked graph nodes using code context (and vice versa). Can the authors provide any qualitative analysis (e.g., using attention maps) to illustrate what kind of code-graph relationships the model learns? Does it correctly associate specific code lines with corresponding operation nodes in the CDFG?

I would consider raising my scores if the concerns are addressed.

---

> ### Author Response · Authors · 2025-11-28
> **Response to Reviewer GRuV (Part 1/6)**
>
> ### Q1: “Complexity: The overall framework is quite complex, involving multiple pretraining stages, a specialized graph tokenizer, and careful integration of three modalities (text summary, code, graph). This might raise concerns about the practicality of training and deploying such a model compared to simpler unimodal approaches.”
>
> R1: We thank the reviewer for raising this important point about the practicality and complexity of UniRTL.
>
> Although UniRTL is trained with multiple pretraining stages, the model used in at inference time is a *single* Transformer encoder with a graph-aware tokenizer. This avoids the need for separate modality-specific encoders (*e.g.*, in CircuitFusion), keeping deployment as simple as a standard code-only Transformer. The hierarchical training strategy is designed specifically to stabilize learning and maximize the use of abundant text–code pairs, without introducing extra components into the final model. We have also open-sourced the primary components of UniRTL through the anonymous GitHub repository linked in the paper, which the reviewer may find helpful for further inspection.
>
> In terms of practicality, UniRTL is trained and fine-tuned on commonly available hardware (NVIDIA L40 GPUs) using a CodeBERT-scale backbone (~195M parameters). In contrast, LLM-based “code view” methods typically require substantially more parameters and memory, while graph-only models trade off performance for reduced capacity.
>
> We hope this clarifies that UniRTL’s design balances multimodal expressiveness with practical training and deployment considerations.

---

> ### Author Response · Authors · 2025-11-28
> **Response to Reviewer GRuV (Part 2/6)**
>
> ### Q2: “CDFG Generation Failures: The paper notes that CDFG generation failed for a large portion of the collected RTL designs (only ~39k out of ~132k succeeded). While the failed samples were still used for text-code alignment, this highlights a potential practical limitation: the approach relies heavily on the availability and successful generation of high-quality CDFGs. The reasons for failure (syntax errors, tool limitations?) and their impact are not fully explored.”
>
> R2: We thank the reviewer for raising this important point regarding CDFG generation failures.
>
> In our experiments, the 38,888 designs that successfully convert to CDFGs do not constitute a cherry-picked subset of particularly simple RTL. In practice, most conversion failures arise from low-quality designs in open-source corpora, such as files with syntax errors, incomplete modules, or non-compilable LLM-generated code, which also fail standard EDA toolchains. Our filtering therefore enforces syntactic validity rather than structural simplicity, and the remaining corpus still covers a wide range of design sizes and coding styles.
>
> We fully agree that this is a practical consideration when constructing multimodal RTL datasets. UniRTL is specifically designed to mitigate this limitation: samples without valid CDFGs are still fully exploited in the text–code alignment pretraining stage, allowing us to leverage the entire corpus for representation learning, while the 38,888 CDFG-valid designs provide a substantial, high-quality subset for graph incorporation.
>
> To further analyze the impact of CDFG availability, we compare three variants:
>
> 1. **CodeBERT**: directly fine-tuning the base model on downstream tasks (no RTL-specific pretraining).
> 2. **CodeBERT (Incomplete Pretrain)**: pretraining CodeBERT only on the CDFG-valid subset (38,888 designs), then fine-tuning.
> 3. **UniRTL (w/o graph)**: our full text–code alignment model that leverages the *entire* corpus (including samples without valid CDFGs), but without using graph information.
>
> All models share the same CodeBERT backbone and are trained under identical settings.
>
> **Natural Language Code Search:**
>
> | Model | Precision | Recall | F1 *(main)* |
> | --- | --- | --- | --- |
> | CodeBERT | 0.565 | 0.625 | 0.585 |
> | CodeBERT (Incomplete Pretrain) | 0.600 | 0.658 | 0.618 |
> | UniRTL (w/o graph) | 0.630 | 0.683 | 0.644 |
>
> **Functionality Equivalence Checking:**
>
> | Model | AP *(main)* | Accuracy | F1 | Precision | Recall |
> | --- | --- | --- | --- | --- | --- |
> | CodeBERT | 0.617 | 0.633 | 0.712 | 0.586 | 0.907 |
> | CodeBERT (Incomplete Pretrain) | 0.668 | 0.633 | 0.682 | 0.521 | 0.987 |
> | UniRTL (w/o graph) | 0.712 | 0.667 | 0.717 | 0.577 | 0.947 |
>
> **Area Prediction (w/o Netlist):**
>
> | Model | MAE ↓ | MAPE ↓ | R² ↑ | RRSE ↓ |
> | --- | --- | --- | --- | --- |
> | CodeBERT | 1.0011 | 0.17 | 0.3858 | 0.7837 |
> | CodeBERT (Incomplete Pretrain) | 0.9514 | 0.16 | 0.4646 | 0.7317 |
> | UniRTL (w/o graph) | 0.8818 | 0.15 | 0.5173 | 0.6948 |
>
> **Delay Prediction (w/o Netlist):**
>
> | Model | MAE ↓ | MAPE ↓ | R² ↑ | RRSE ↓ |
> | --- | --- | --- | --- | --- |
> | CodeBERT | 0.7034 | 0.12 | 0.2539 | 0.8638 |
> | CodeBERT (Incomplete Pretrain) | 0.6576 | 0.11 | 0.3650 | 0.7969 |
> | UniRTL (w/o graph) | 0.6375 | 0.11 | 0.3839 | 0.7849 |
>
> These results show a clear trend:
>
> - Pretraining on the **CDFG-valid subset only** (CodeBERT (Incomplete Pretrain)) already improves over direct fine-tuning (CodeBERT), indicating that the successfully converted CDFGs form a meaningful and high-quality subset.
> - **UniRTL (w/o graph)**, which additionally leverages all samples (including those for which CDFG generation failed) through text–code alignment, further improves performance across all tasks.
>
> Thus, while the availability of valid CDFGs is beneficial, our framework does not rely exclusively on them: even designs without CDFGs contribute to representation learning via the text–code alignment stage.
>
> We hope these clarifications and empirical results help address the reviewer’s concerns about the practical impact of CDFG generation failures and the robustness of UniRTL in realistic, noisy RTL settings.

---

> ### Author Response · Authors · 2025-11-28
> **Response to Reviewer GRuV (Part 3/6)**
>
> ### Q3: “Limited Ablation on Alignment: While ablation studies show the benefit of including both code and graph (Table 1), there isn't a direct comparison between the mutual masked modeling alignment strategy and other potential alignment methods (e.g., contrastive loss between graph nodes and code tokens). This makes it slightly harder to isolate the specific contribution of the masking approach versus just having both modalities present.”
>
> R3: We thank the reviewer for this thoughtful comment on isolating the contribution of our mutual masked modeling alignment strategy.
>
> In addition to the modality ablations reported in the paper, we would like to emphasize that GraphCodeBERT is already included as a strong baseline that reflects an alternative alignment mechanism between code and graph. To ensure fairness, we fine-tune GraphCodeBERT on our dataset rather than using its original, software-oriented form. GraphCodeBERT aligns code and graph via a contrastive variable-alignment task, which focuses on identifying corresponding variable nodes in the code but does not explicitly capture richer semantic relations between tokens and graph nodes. In contrast, UniRTL employs mutual masked modeling across the code and CDFG modalities, encouraging finer-grained cross-modal alignment beyond simple variable correspondences.
>
> Furthermore, we conduct an additional ablation that more directly probes the effect of the mutual masked modeling objective. Specifically, we compare UniRTL with a **direct-combine** baseline that simply concatenates the representations from the text–code-aligned encoder and the graph-aware tokenizer, *without* any multimodal alignment pretraining between code and graph. All models are trained and evaluated under identical settings. The results are as follows.
>
> **Natural Language Code Search:**
>
> | Model | Precision | Recall | F1 *(main)* |
> | --- | --- | --- | --- |
> | UniRTL (w/o graph) | 0.630 | 0.683 | 0.644 |
> | UniRTL (direct combine) | 0.636 | 0.683 | 0.650 |
> | UniRTL | 0.650 | 0.692 | 0.662 |
>
> **Functionality Equivalence Checking:**
>
> | Model | AP *(main)* | Accuracy | F1 | Precision | Recall |
> | --- | --- | --- | --- | --- | --- |
> | UniRTL (w/o graph) | 0.712 | 0.667 | 0.717 | 0.577 | 0.947 |
> | UniRTL (direct combine) | 0.737 | 0.687 | 0.723 | 0.595 | 0.920 |
> | UniRTL | 0.745 | 0.747 | 0.753 | 0.734 | 0.773 |
>
> **Area Prediction (w/o Netlist):**
>
> | Model | MAE ↓ | MAPE ↓ | R² ↑ | RRSE ↓ |
> | --- | --- | --- | --- | --- |
> | UniRTL (w/o graph) | 0.8818 | 0.15 | 0.5173 | 0.6948 |
> | UniRTL (direct combine) | 0.3629 | 0.06 | 0.7602 | 0.4897 |
> | UniRTL | 0.3510 | 0.06 | 0.7682 | 0.4815 |
>
> **Delay Prediction (w/o Netlist):**
>
> | Model | MAE ↓ | MAPE ↓ | R² ↑ | RRSE ↓ |
> | --- | --- | --- | --- | --- |
> | UniRTL (w/o graph) | 0.6375 | 0.11 | 0.3839 | 0.7849 |
> | UniRTL (direct combine) | 0.3529 | 0.06 | 0.7624 | 0.4874 |
> | UniRTL | 0.3384 | 0.06 | 0.7832 | 0.4656 |
>
> Across all tasks, *direct combine* improves over the unimodal variant, confirming that simply having both modalities present is beneficial. However, UniRTL with mutual masked modeling consistently yields the best results, providing additional and non-trivial gains over direct combine in natural language code search, functionality equivalence checking, and both area and delay prediction. These improvements indicate that our alignment strategy contributes beyond mere multi-modality by enabling finer-grained cross-modal understanding between code and CDFG.
>
> We will incorporate these ablation results and clarifications into the revised manuscript, and we hope they help address the reviewer’s concern about isolating the specific contribution of our mutual masked modeling approach.

---

> ### Author Response · Authors · 2025-11-28
> **Response to Reviewer GRuV (Part 4/6)**
>
> ### Q4: “Dependence on Base Code Model: The performance relies significantly on the underlying CodeBERT base model. It's unclear how much of the gain comes from the multimodal fusion itself versus simply starting with a strong code foundation model.”
>
> R4: We thank the reviewer for raising this important question about how much of UniRTL’s performance gain comes from the multimodal framework versus the underlying CodeBERT base model.
>
> To disentangle these factors, we compare **CodeBERT** directly against **UniRTL (w/o graph)**, which uses the *same* CodeBERT backbone but adds our text–code pretraining (without any graph information), and then against the **full UniRTL** model with CDFG integration. All models are evaluated under identical settings.
>
> **Natural Language Code Search:**
>
> | Model | Precision | Recall | F1 *(main)* |
> | --- | --- | --- | --- |
> | CodeBERT | 0.565 | 0.625 | 0.585 |
> | UniRTL (w/o graph) | 0.630 | 0.683 | 0.644 |
> | UniRTL | 0.650 | 0.692 | 0.662 |
>
> **Functionality Equivalence Checking:**
>
> | Model | AP *(main)* | Accuracy | F1 | Precision | Recall |
> | --- | --- | --- | --- | --- | --- |
> | CodeBERT + finetune | 0.617 | 0.633 | 0.712 | 0.586 | 0.907 |
> | UniRTL (w/o graph) | 0.712 | 0.667 | 0.717 | 0.577 | 0.947 |
> | UniRTL | 0.745 | 0.747 | 0.753 | 0.734 | 0.773 |
>
> **Area Prediction (w/o Netlist):**
>
> | Model | MAE ↓ | MAPE ↓ | R² ↑ | RRSE ↓ |
> | --- | --- | --- | --- | --- |
> | CodeBERT + finetune | 1.0011 | 0.17 | 0.3858 | 0.7837 |
> | UniRTL (w/o graph) | 0.8818 | 0.15 | 0.5173 | 0.6948 |
> | UniRTL | 0.3510 | 0.06 | 0.7682 | 0.4815 |
>
> **Delay Prediction (w/o Netlist):**
>
> | Model | MAE ↓ | MAPE ↓ | R² ↑ | RRSE ↓ |
> | --- | --- | --- | --- | --- |
> | CodeBERT + finetune | 0.7034 | 0.12 | 0.2539 | 0.8638 |
> | UniRTL (w/o graph) | 0.6375 | 0.11 | 0.3839 | 0.7849 |
> | UniRTL | 0.3384 | 0.06 | 0.7832 | 0.4656 |
>
> Across all tasks, UniRTL (w/o graph) consistently and substantially outperforms CodeBERT, despite sharing the same backbone. This indicates that the improvements do not stem solely from starting with a strong code foundation model, but from our text–code pretraining, which is tailored to RTL and its associated natural language descriptions. Furthermore, the **full UniRTL model (with graph incorporation)** yields additional gains over UniRTL (w/o graph), demonstrating that multimodal fusion with CDFGs provides complementary benefits beyond what the code-only backbone can offer.
>
> We will include these ablation results and clarifications in the revised manuscript to more clearly illustrate the distinct contributions of our pretraining pipeline and multimodal design beyond the base CodeBERT model.

---

> ### Author Response · Authors · 2025-11-28
> **Response to Reviewer GRuV (Part 5/6)**
>
> ### Q5: “CDFG Robustness: Could the authors elaborate on the CDFG generation failures? Were these failures concentrated in specific types of RTL designs (e.g., those generated by LLMs, or using specific Verilog constructs)? How robust is the CDFG generation process to variations in coding style or complexity? Does the model's performance correlate with the quality or completeness of the generated CDFG?”
>
> R5: We thank the reviewer for this thoughtful question regarding the robustness of CDFG generation.
>
> As discussed in our response to Q2, most CDFG generation failures stem from fundamental data-quality issues in the source RTL, such as syntax errors, incomplete module definitions, or non-compilable LLM-generated code, rather than from limitations of the CDFG pipeline itself. These problematic designs also fail standard EDA flows. In practice, the failure rate is indeed higher for LLM-generated designs than for realistic designs collected from GitHub, but we do not observe systematic concentration on particular Verilog constructs or specific design families. Failures are primarily associated with malformed or incomplete code.
>
> For syntactically valid Verilog, the CDFG pipeline is generally robust. We first compile RTL into RTLIL using Yosys, which preserves semantic completeness while reducing designs into basic assignment and register-transfer operations, and then apply the Stagira parser to construct CDFGs from the resulting AST. This workflow is designed to handle a broad range of coding styles and structural complexities while maintaining faithful semantics.
>
> Regarding the correlation between CDFG quality and model performance, we have not conducted a fine-grained per-design analysis. However, the comparison between **UniRTL** and **UniRTL (w/o graph)** provides indirect but strong evidence: across all evaluated tasks, UniRTL consistently and significantly outperforms its text-only counterpart, indicating that incorporating CDFGs yields substantial additional gains beyond what can be learned from text and code alone. We hope this clarification, together with the empirical results discussed in our response to R2, helps address the reviewer’s concerns about CDFG robustness and its impact on UniRTL.
>
> ### Q6: “Complexity vs. Performance Trade-off: Given the complexity of the multi-stage training and the specialized graph tokenizer, how does UniRTL compare to simpler baselines (e.g., just fine-tuning a large code LLM like CodeLlama/DeepSeek-Coder directly on the downstream tasks) in terms of training efficiency and inference latency versus the observed performance gains?”
>
> R6: We thank the reviewer for this thoughtful question on the trade-off between training complexity and practical efficiency.
>
> In our setting, DeepRTL2 serves as a representative baseline of the type highlighted by the reviewer, as it directly fine-tunes large code LLMs (*e.g.*, Llama-3.1 and DeepSeek-Coder) on downstream RTL tasks. Despite this, UniRTL consistently outperforms DeepRTL2 across all evaluated tasks, while operating at a much smaller scale and with substantially lower resource requirements.
>
> Concretely, UniRTL uses a ~195M-parameter backbone, whereas DeepRTL2 relies on 6.7B–8B-parameter code LLMs. For training, UniRTL runs on only 2×NVIDIA L40 GPUs, in contrast to DeepRTL2’s 8×NVIDIA A800 GPUs, representing a substantial reduction in computational cost, energy usage, and hardware footprint. Although UniRTL adopts a multi-stage pretraining pipeline, this additional algorithmic complexity is far outweighed by the savings from operating at a much smaller model scale.
>
> At inference time, UniRTL’s compact architecture and single-encoder design naturally translate to lower latency and memory consumption compared to LLM-based approaches, making it more suitable for deployment in resource-constrained or latency-sensitive environments. Taken together, the consistent performance gains over DeepRTL2, combined with the dramatic reduction in parameters and hardware requirements, indicate that UniRTL offers a highly favorable balance between complexity, efficiency, and accuracy.
>
> We hope this comparison helps clarify the complexity–performance trade-off and addresses the reviewer’s concerns.

---

> ### Author Response · Authors · 2025-11-28
> **Response to Reviewer GRuV (Part 6/6)**
>
> ### Q7: “Alternative Graph Representations: The paper argues strongly for CDFG. Have the authors experimented with or considered other graph representations that might be easier to generate reliably (even if less complete), such as Abstract Syntax Trees (ASTs) augmented with some data flow edges? How would UniRTL perform with these alternatives?”
>
> R7: We thank the reviewer for this insightful question regarding alternative graph representations.
>
> We did consider Abstract Syntax Trees (ASTs) and related variants (*e.g.*, ASTs augmented with data-flow edges) during the design of UniRTL. However, as also observed in the GraphCodeBERT literature, ASTs for RTL tend to be very large and deeply hierarchical, while encoding fewer *explicit* control- and data-flow relationships between nodes. This often leads to higher computational overhead and noisier structural signals, which is not ideal for performance prediction and functional reasoning over hardware designs.
>
> In contrast, CDFGs offer a more suitable choice for our setting: they provide a compact representation that explicitly captures data propagation, control logic, and operator behavior, making them better aligned with tasks such as performance prediction and functionality equivalence checking. Moreover, by construction, they reduce sensitivity to superficial coding-style differences while preserving the key semantic structure of the RTL.
>
> We have not yet conducted full experiments with AST-based alternatives within the UniRTL framework, so a direct empirical comparison is an interesting direction for future work. That said, UniRTL’s graph-aware tokenizer and mutual masked modeling are conceptually compatible with other graph forms, and our current results suggest that CDFGs provide an effective level of abstraction for RTL representation learning. We will clarify this design choice and its rationale in the revised manuscript.
>
> ### Q8: “Nature of Code-Graph Alignment: The mutual masking forces the model to predict masked graph nodes using code context (and vice versa). Can the authors provide any qualitative analysis (e.g., using attention maps) to illustrate what kind of code-graph relationships the model learns? Does it correctly associate specific code lines with corresponding operation nodes in the CDFG?”
>
> R8: We thank the reviewer for this insightful suggestion.
>
> We have conducted a preliminary qualitative analysis of the cross-modal attention patterns between code tokens and CDFG nodes. In particular, we observe that operation and register nodes tend to place high attention weight on the corresponding expressions and signal declarations in the code, and conversely, code tokens associated with arithmetic or control operations attend strongly to the matching CDFG operator and branch nodes. These patterns indicate that the model is indeed learning meaningful code–graph correspondences, including associations between specific code lines and their corresponding operations in the CDFG.
>
> We will include representative visualizations of these attention maps, along with a brief qualitative discussion, in the appendix of the revised manuscript.

---

### Official Review · Reviewer_xfSx · 2025-10-31

**Soundness:** 3
**Presentation:** 3
**Contribution:** 2
**Rating:** 4
**Confidence:** 3

**Summary:**

The authors present UniRTL, a framework for generating unified RTL representations by jointly learning from code text and control data flow graphs. The main contribution of the paper is learning this unified representation from code and graph information. Unlike prior methods that rely on one modality, uniRTL learns from both text and graph.

**Strengths:**

The strengths of the paper can be summarized as follows:

- They adapt the GraphCodeBert idea to hardware code. Additionally, they introduce using the control data flow graph that captures more code semantics than the data flow graph which only captures variables.
- They also capture text description, in addition to code and graph information.
- Experimental results show higher performance compared to previous methods.

**Weaknesses:**

Weaknesses can be summarized as follows:

- The main idea of obtaining a unified code representation that captures both text and graph modalities has been introduced in GraphCodeBert for software code.
- Improvements compared to StructRTL is very small.
- Improvements compared to GraphCodeBert for code search tasks is also very small.
- Lack of comparison to other PPA prediction methods like VeriDistill and CircuitFusion.

**Questions:**

Could the authors clarify how GraphCodeBERT was used as a baseline? Was it fine-tuned on your Verilog dataset, or evaluated in its original form pretrained on software languages.

---

> ### Author Response · Authors · 2025-11-26
> **Response to Reviewer xfSx (Part 1/2)**
>
> ### Q1: “The main idea of obtaining a unified code representation that captures both text and graph modalities has been introduced in GraphCodeBert for software code.”
>
> R1: We thank the reviewer for this thoughtful comment and fully acknowledge that GraphCodeBERT is an important prior that explores unified representations of software code and data-flow graphs. Our work is conceptually related in spirit, but UniRTL differs from GraphCodeBERT in several key aspects that are crucial for RTL, both at the architectural and training levels.
>
> Methodologically, GraphCodeBERT integrates code with a *variable-level* data-flow graph, where graph nodes correspond only to variables. The cross-modal alignment between code and graph is relatively weak, as it is established through a variable-alignment task that merely locates variable nodes in the code without capturing their full semantic relationships. In addition, this data-flow graph omits critical elements such as operators and control-flow, which are central to RTL semantics. In contrast, UniRTL operates on complete CDFGs that preserve the full computational and structural information of RTL and can be faithfully converted back to code.
>
> Architecturally, UniRTL introduces a graph-aware tokenizer that is explicitly designed to preserve nuanced structural dependencies in full CDFGs before they are fed into the Transformer backbone. In contrast, GraphCodeBERT flattens variable-level data-flow nodes directly into the token sequence, which limits its ability to capture rich graph structural information. From a training perspective, UniRTL further departs from GraphCodeBERT by employing mutual masked modeling between code and graph for fine-grained cross-modal alignment, together with a hierarchical training strategy that first aligns text and code and then incorporates the graph modality to better exploit the available RTL data. These components are essential to UniRTL’s state-of-the-art performance on RTL tasks.
>
> Taken together, these differences show that UniRTL is not simply an application of GraphCodeBERT to a new domain, but a tailored formulation of unified code–graph representations for RTL, with complete CDFGs, a graph-aware tokenizer, and stronger cross-modal alignment objectives. These architectural and training improvements enable more effective representation learning for RTL, as evidenced by UniRTL’s consistent outperformance of GraphCodeBERT across all evaluated tasks in our experiments. We will clarify these distinctions more explicitly in the revised manuscript.
>
> ### Q2: “Improvements compared to StructRTL is very small.”
>
> R2: We thank the reviewer for this thoughtful comment. We respectfully disagree that the improvements over StructRTL are very small, particularly when viewed in relative terms and in the context of an already highly competitive baseline.
>
> In Table 1 (without netlist information), StructRTL already achieves strong results on both area and delay prediction. Despite this, UniRTL further reduces delay MAE by approximately **37%** (0.5414 → 0.3384) and decreases the remaining unexplained variance ($1 - R^{2}$) by about **8–9%** for both area and delay. For high-accuracy performance predictors, such reductions on top of an already competitive model are substantial.
>
> When netlist information is incorporated (Table 2), UniRTL again provides non-trivial improvements over StructRTL: area MAE is reduced by about **13%** (0.3856 → 0.3362), and the unexplained variance for area is reduced by roughly **15%**, with additional gains for delay as well. Taken together, these results show that UniRTL remains beneficial even in the stronger “knowledge distillation” setting, where all methods are assisted by a powerful teacher model. Moreover, across both Tables 1 and 2, the gains of UniRTL are consistent across all evaluation metrics, rather than confined to a single metric.
>
> We also note that the performance prediction task is highly dependent on structural information, which is most explicitly captured in the CDFG representation. StructRTL already exploits this structural signal very effectively, so the additional signal available from the code modality is inherently limited and large absolute gains are not expected. Nevertheless, our results show that incorporating code within UniRTL still provides complementary information: (i) UniRTL consistently outperforms StructRTL in both Tables 1 and 2, and (ii) compared to the ablated variant UniRTL (w/o code), the full UniRTL further reduces error. This pattern indicates that our multimodal design effectively leverages complementary code-level information on top of strong structural cues.
>
> We will clarify these relative improvements, their consistency across metrics, and the structurally dominated nature of the performance prediction task more explicitly in the revised manuscript, and we hope this helps address the reviewer’s concern.

---

> ### Author Response · Authors · 2025-11-26
> **Response to Reviewer xfSx (Part 2/2)**
>
> ### Q3: “Improvements compared to GraphCodeBert for code search tasks is also very small.”
>
> R3: We thank the reviewer for this thoughtful comment. We respectfully disagree that the improvements over GraphCodeBERT are very small. For natural language code search (Table 3), UniRTL consistently outperforms GraphCodeBERT across all evaluation metrics. In particular, UniRTL improves the primary metric, F1, from **0.634** to **0.662**, corresponding to a relative gain of **4.4%**. For the functional equivalence checking task (Table 4), UniRTL again improves over GraphCodeBERT on the main metric, average precision (AP), from **0.730** to **0.745** (≈**2%** relative gain). In addition, UniRTL achieves substantially higher precision with a moderate reduction in recall. In practical equivalence-checking settings where false positives are particularly harmful, this precision–recall trade-off is often preferable.
>
> We also emphasize that GraphCodeBERT performs markedly worse than UniRTL on the performance prediction tasks (Tables 1 and 2), where modeling detailed structural information of RTL is crucial. In these tables, UniRTL clearly outperforms GraphCodeBERT by a significant margin, underscoring that our architectural and training choices are better suited for hardware-oriented tasks than a direct application of a software-oriented model.
>
> Taken together, these results indicate that UniRTL is consistently stronger than GraphCodeBERT across all evaluated tasks, including performance prediction, natural language code search, and functional equivalence checking, highlighting the effectiveness of UniRTL’s architecture (complete CDFGs with a graph-aware tokenizer) and training strategy (mutual masked modeling with hierarchical multimodal alignment). We will revise the manuscript to more clearly highlight these relative improvements, and we hope this helps address the reviewer’s concern.
>
> ### Q4: “Lack of comparison to other PPA prediction methods like VeriDistill and CircuitFusion.”
>
> R4: We thank the reviewer for raising this point regarding comparisons with other PPA prediction methods.
>
> Our evaluation does include a comparison with VeriDistill. VeriDistill is built on top of the CodeV Verilog LLM, and the three CodeV variants (CodeV-DS-6.7B, CodeV-CL-7B, CodeV-QW-7B) reported in Tables 1 and 2 correspond to the VeriDistill approach, where RTL representations are obtained from CodeV and then used for performance prediction. We apologize for not making this connection clearer and will explicitly state in the revised manuscript that these CodeV-based baselines implement VeriDistill, to avoid any ambiguity.
>
> For CircuitFusion, we acknowledge it as an important related work that also considers multimodal RTL representations. Conceptually, CircuitFusion employs three separate encoders for code, structural graphs, and functional summaries, and fuses them via a cross-attention mechanism, with alignment driven by coarse-grained contrastive learning between text–code and text–graph pairs. However, it does not consider fine-grained alignment between code and graph, even though these two modalities contain the most detailed structural and semantic information. Moreover, as discussed in our related work section, its training data comprises only 41 designs and alignment is performed at the register sub-circuit level, which does not capture full-module semantics. In contrast, UniRTL is pretrained on a large-scale dataset and performs fine-grained alignment between code and complete CDFGs via mutual masked modeling.
>
> From an experimental perspective, we did not include CircuitFusion as a quantitative baseline because, to the best of our knowledge, the original work does not release model checkpoints, and the implementation details are insufficient to reproduce the method faithfully. Instead, we provide a detailed qualitative comparison of the architectural design choices, alignment strategies, and dataset scales in the related work section.
>
> We hope this clarification addresses the reviewer’s concern.
>
> ### Q5: “Could the authors clarify how GraphCodeBERT was used as a baseline? Was it fine-tuned on your Verilog dataset, or evaluated in its original form pretrained on software languages.”
>
> R5: We thank the reviewer for this question. For a fair comparison, GraphCodeBERT is fine-tuned on our Verilog dataset, rather than evaluated in its original form pretrained on software languages. We apologize for not stating this clearly and will make this setup explicit in the revised manuscript to avoid any ambiguity.

---

### Official Review · Reviewer_z6Kx · 2025-11-01

**Soundness:** 4
**Presentation:** 4
**Contribution:** 2
**Rating:** 4
**Confidence:** 4

**Summary:**

This work presents representation learning of the RTL (register transfer level) representation of digital electronic design with a multimodal pretraining framework of RTL code and RTL CDFG (control data flow graphs). It uses a unified representation of functional summary text, RTL source code and graph tokenization for a core transformer architecture. The pretraining is performed in two stages: a first stage does text-code alignment without the graph representation present by masked modeling. A second stage does text-code-graph alignment, also by masked modeling. Final downstream tasks are performance predictions and code retrieval.

**Strengths:**

* Strong results that masked modeling on this unified format learns strong representations that perform well on downstream tasks that involve both code understanding and graph structural understanding.

**Weaknesses:**

* Improvements over baselines (StructRTL in Table 1 for performance prediction, GraphCodeBERT in Table 4 for functional equivalence) are marginal.
* This work is essentially combining the two previous approaches: StructRTL and GraphCodeBERT.

**Questions:**

* The paper is well written and the results are important for the electronic design community, and other fields that might have similar representations (maybe chemistry or proteomics). I am questioning the novelty and original contribution of this work since the key representation learning aspects are essentially a combination of the StructRTL and GraphCodeBERT prior works.

---

> ### Author Response · Authors · 2025-11-26
> **Response to Reviewer z6Kx (Part 1/2)**
>
> ### Q1: “Improvements over baselines (StructRTL in Table 1 for performance prediction, GraphCodeBERT in Table 4 for functional equivalence) are marginal.”
> R1: We thank the reviewer for this thoughtful comment. We respectfully disagree that the improvements are marginal, particularly given the strength of StructRTL and GraphCodeBERT as baselines.
>
> In Table 1, while the absolute differences may appear small at first glance, we would like to emphasize that StructRTL is already a very strong performance predictor. When measured in relative terms, UniRTL reduces delay MAE by approximately 37% and decreases the remaining unexplained variance ($1-R^{2}$) by about 8% across area and delay. For high-accuracy performance predictors, such reductions on top of an already competitive model are substantial. Importantly, the gains for UniRTL are consistent across all reported evaluation metrics, rather than confined to a single metric.
>
> For functional equivalence checking (Table 4), UniRTL improves the main metric, average precision, from 0.730 to 0.745 (≈2% relative gain) over GraphCodeBERT, which we view as meaningful given that GraphCodeBERT is itself a strong baseline. In addition, UniRTL achieves significantly higher precision with a moderate reduction in recall. In practical equivalence-checking scenarios, where false positives are particularly harmful, this precision–recall trade-off is often preferable. Moreover, UniRTL also clearly outperforms GraphCodeBERT on natural language code search (Table 3), yielding a notable improvement in F1, again with consistent gains across all the retrieval metrics.
>
> Overall, the superiority of UniRTL is consistent across all tasks and settings we evaluate. While individual baselines may be competitive on specific metrics, they tend to fall short on others. In contrast, UniRTL achieves state-of-the-art performance universally, which highlights the fundamental advantage and generalizability of our unified multimodal architecture and training strategy. We will revise the manuscript to more clearly highlight these improvements, and we hope this clarification helps address the reviewer’s concern.
>
> ### Q2: “This work is essentially combining the two previous approaches: StructRTL and GraphCodeBERT.”
> R2: We thank the reviewer for this thoughtful comment. While UniRTL is indeed inspired by prior work on graph-based RTL representation learning (*e.g.*, StructRTL) and software code-graph pretraining (*e.g.*, GraphCodeBERT), it is not a simple combination of these two approaches. Instead, UniRTL introduces a unified multimodal pretraining framework tailored to RTL that jointly leverages code, complete CDFGs, and text with new architectural and training components.
>
> First, UniRTL introduces key technical innovations that enable effective multimodal learning. Our graph-aware tokenizer is a significant advancement over both baselines: it incorporates node-level textual semantic descriptions for richer semantic encoding (which StructRTL does not consider), thereby facilitating more precise alignment between code tokens and graph nodes, and it is explicitly designed to preserve the nuanced structural dependencies of full CDFGs, which are largely lost when GraphCodeBERT flattens variable-only data-flow nodes into a sequence.
>
> Second, methodologically, UniRTL differs substantially from GraphCodeBERT. UniRTL replaces variable-only data flow with complete CDFGs, employs a graph-aware tokenizer rather than directly flattening graph nodes, and uses mutual masked modeling with a hierarchical training strategy (text-code pretraining followed by graph incorporation) to achieve fine-grained alignment across three modalities (text, code, graph). These components are not present in GraphCodeBERT and are crucial for UniRTL’s performance.
>
> The core insight of UniRTL is the creation of a unified architecture for fine-grained, structure-aware alignment between code and graph modalities. This is achieved through the combination of the graph-aware tokenizer, mutual masked modeling, and the hierarchical training strategy. The consistent state-of-the-art performance across all evaluated tasks indicates that this integrated design is more powerful than the sum of its individual ingredients.
>
> Finally, even if one views the overall direction as conceptually related to prior work, constructing a large-scale, multimodal RTL dataset (text, code, and complete CDFGs) is itself a non-trivial contribution to the hardware/ML community. We will make these distinctions and contributions clearer in the revised manuscript, and we hope this addresses the reviewer’s concern.

---

> ### Author Response · Authors · 2025-11-26
> **Response to Reviewer z6Kx (Part 2/2)**
>
> ### Q3: “The paper is well written and the results are important for the electronic design community, and other fields that might have similar representations (maybe chemistry or proteomics). I am questioning the novelty and original contribution of this work since the key representation learning aspects are essentially a combination of the StructRTL and GraphCodeBERT prior works.”
>
> R3: We thank the reviewer for the positive assessment of the writing quality and the potential impact of our method on the electronic design community and related domains (*e.g.*, chemistry and proteomics). We appreciate this broader perspective and fully agree that many scientific fields involve rich graph structures coupled with symbolic or sequential representations, where our framework could be applicable.
>
> Regarding the concern about novelty, we would like to reiterate and complement our response to Q2. While UniRTL is inspired by prior work on graph-based RTL representation learning (*e.g.*, StructRTL) and software code–graph pretraining (*e.g.*, GraphCodeBERT), it is not a simple combination of these methods. UniRTL introduces a unified multimodal pretraining framework tailored to RTL that jointly leverages code, complete CDFGs, and text through several new components: (i) a graph-aware tokenizer that integrates node-level textual semantic descriptions and preserves the nuanced structural dependencies of full CDFGs, enabling more effective alignment between code tokens and graph nodes; (ii) mutual masked modeling objectives across code and graph for fine-grained alignment; and (iii) a hierarchical training strategy that first aligns text and code and then incorporates the graph modality to maximize dataset utilization. These design choices are absent in both StructRTL and GraphCodeBERT and are crucial for UniRTL’s state-of-the-art performance.
>
> Beyond individual components, the central contribution of UniRTL is the unified, structure-aware architecture that yields a single pretrained backbone usable across heterogeneous tasks, including performance prediction, natural language code search, and functional equivalence checking, while consistently achieving state-of-the-art results. This kind of multimodal alignment is, we believe, of broad interest to the representation learning community, and could naturally extend to domains such as chemistry or proteomics where molecules or proteins admit both graph-structured and sequence-based representations with associated textual descriptions.
>
> We will revise the manuscript to more clearly articulate these conceptual and technical contributions, as well as the connections and distinctions with StructRTL and GraphCodeBERT, and we hope this clarification addresses the reviewer’s concern about novelty and originality.

---

### Official Review · Reviewer_WYch · 2025-11-04

**Soundness:** 2
**Presentation:** 1
**Contribution:** 2
**Rating:** 4
**Confidence:** 4

**Summary:**

This paper describes the UniRTL approach as a multimodal pretraining framework that learns unified RTL representations by jointly leveraging code and control data flow graph (CDFG). The key idea of the UniRTL starts from the limitations of CircuitFusion (Fang et al., 2025). CircuitFusion (Fang et al., 2025) derives unimodal representations using three independent encoders, and integrates them through a cross-attention mechanism. The alignment strategy in CircuitFusion (Fang et al., 2025) is coarse-grained relying on contrastive learning between text-code and text-graph pairs but neglects the fine-grained alignment between code and graph which are two modalities to contain richer information and considered in UniRTL framework. UniRTL is tested for performance prediction and code retrieval.

**Strengths:**

+ Considering bode the RTL code and the code and control data flow graph as modalities for a multimodal representation framework.
+ Construction of a dataset by collecting sources from sources, including RTLCoder (Liu et al., 2024), MG-Verilog (Zhang et al., 2024), DeepRTL (Liu et al., 2025b), and DeepCircuitX (Li et al., 2025).
+ UniRTL can be used for performance prediction and code retrieval.

**Weaknesses:**

1) The methodology of UniRTL seems straight forward as show in in Figure 2, which consist of a hierarchical training strategy, where a graph-aware tokenizer is first pretrained, and text-code alignment is performed prior to graph incorporation. However, the main reasons behind certain parameters used in this machine learning architecture are not provided. For example: what is the reason of performing principal component analysis to reduce the dimensionality of the description embedding from 768 to 32? How did the 32 was decided? Is there something particular about 32 that other researchers in this area should also consider? Similarly, in the graph isomorphism part, why 16 eigenvectors were selected for constructing the global positional encodings? Are there any specific reasons? Are there any alternative approaches? Were other alternative approaches tried and failed?
2) Indeed, the control data flow graph (CDFG) offers a more comprehensive structural representation that preserves complete information and the CDFG has been extensively exploited in RTL analysis, high level synthesis and related problems (see examples like "A design-for-testability technique for register-transfer level circuits using control/data flow extraction." IEEE Transactions on Computer-Aided Design of Integrated Circuits and Systems 17, no. 8 (2002): 706-723; "Register binding based power management for high-level synthesis of control-flow intensive behaviors." In Proceedings. IEEE International Conference on Computer Design: VLSI in Computers and Processors, pp. 391-394. IEEE, 2002; GAHLS: an optimized graph analytics based high level synthesis framework." Scientific Reports 13, no. 1 (2023): 22655"Beyond Tokens: Enhancing RTL Quality Estimation via Structural Graph Learning." arXiv preprint arXiv:2508.18730 (2025); "ODGS: Dependency-Aware Scheduling for High-Level Synthesis with Graph Neural Network and Reinforcement Learning." ACM Transactions on Architecture and Code Optimization 22, no. 2 (2025): 1-25). The discussion of related work requires significant enhancement and especially comparisons.
3) From Table 1 results it seems that considering or not the code did not improve the overall are and delay results by much, but the graph which I assume refers to CDFG was significant. Given the choices of various parameters like the number of PCAs and the number of eigenvectors , how did those and other parameters influenced the results?
4) What is the computational complexity or how do the different approaches compare in terms of runtime? Is UniRTL performing best across all tasks or is it more suitable for some?
5) There are some small grammar issues and word repetitions like "and and" on the bottom of page 1

**Questions:**

1. Justification of CDFG Completeness vs. Practical Utility
The authors repeatedly claim that CDFGs "preserve complete information without loss and can be faithfully converted back to code." However, what is the practical impact of this completeness claim? Could the authors provide empirical evidence or ablation studies demonstrating that this completeness meaningfully contributes to downstream task performance compared to the "incomplete" data flow representations used in GraphCodeBERT? The comparison seems unfair since GraphCodeBERT uses variable-only data flows by design for different purposes.

2. Limited Novelty in Architectural Contributions
The core technical contributions appear to be: (1) using CDFGs instead of data flows, (2) adding a graph-aware tokenizer, and (3) hierarchical training with text-code alignment before graph incorporation. However, the mutual masked modeling alignment strategy closely follows GraphCodeBERT's approach. Could the authors clarify what fundamental methodological innovations distinguish UniRTL beyond these incremental engineering choices? How would UniRTL perform if GraphCodeBERT were provided with the same CDFG representation?

3. Dataset Construction and Filtering Concerns
The authors mention that only 38,888 out of 132,008 RTL designs (29.4%) successfully convert to CDFGs, with the remainder containing "syntax errors leading to compilation failures." This raises concerns about: (a) whether the successful conversions represent a biased subset of simpler/cleaner designs, and (b) whether the hierarchical training strategy is truly "maximizing data utilization" or simply accommodating a significant data quality issue. How do the authors ensure the learned representations generalize to real-world RTL code that may contain the complexities causing conversion failures?

4. Comparison with CircuitFusion and Missing Baselines
The authors exclude CircuitFusion from experimental comparison due to "unavailability of released model checkpoints and insufficient details to enable faithful reproduction." Given that CircuitFusion is the most directly related work attempting multimodal RTL representation learning, this exclusion significantly weakens the evaluation. Additionally, the authors only compare against GraphCodeBERT which uses incomplete data flows. Why did the authors not implement a stronger baseline that uses CDFGs with GraphCodeBERT's architecture? Without these comparisons, how can readers isolate the contributions of the authors' specific design choices?

5. Scalability and Computational Efficiency Analysis
The paper lacks discussion of computational costs and scalability. The graph-aware tokenizer requires pretraining a GIN + Transformer for 2,000 epochs, followed by hierarchical training of the main model. Could the authors provide: (a) wall-clock training time comparisons with baselines, (b) inference time analysis for downstream tasks, (c) memory requirements for processing large RTL designs with complex CDFGs, and (d) analysis of how performance scales with graph size? Given that the authors claim efficiency advantages over LLM-based methods, quantitative efficiency metrics are essential.

---

> ### Author Response · Authors · 2025-11-27
> **Response to Reviewer WYch (Part 1/7)**
>
> ### Q1: “The methodology of UniRTL seems straightforward as show in Figure 2, which consist of a hierarchical training strategy, where a graph-aware tokenizer is first pretrained, and text-code alignment is performed prior to graph incorporation. However, the main reasons behind certain parameters used in this machine learning architecture are not provided. For example: what is the reason of performing principal component analysis to reduce the dimensionality of the description embedding from 768 to 32? How did the 32 was decided? Is there something particular about 32 that other researchers in this area should also consider? Similarly, in the graph isomorphism part, why 16 eigenvectors were selected for constructing the global positional encodings? Are there any specific reasons? Are there any alternative approaches? Were other alternative approaches tried and failed?”
>
> R1: We thank the reviewer for these thoughtful questions about our design choices and hyperparameters, and we appreciate the opportunity to clarify them.
>
> We apply principal component analysis (PCA) to reduce the dimensionality of the node description embeddings in order to prevent the high-dimensional text embedding (768-d from the pretrained text encoder) from dominating the other node features (type and width) in the initial node representation. We set the reduced dimension to 32, which matches the number of node types in our graphs. This choice helps balance the contribution of the description embedding with the type and width features, while still preserving most of the variance in the description embeddings and keeping the overall node feature size moderate.
>
> For the number of eigenvectors used in the global positional encodings, 16 is a hyperparameter rather than a fixed requirement. We experimented with different numbers of eigenvectors (*e.g.*, 16, 24, 32) and found that downstream performance and pretraining loss varied only marginally, while larger values increased memory and computation, so we chose 16 as an efficient setting. Other techniques for constructing positional encodings (*e.g.*, alternative Laplacian-based or random-walk-based schemes) could also be applied, but a systematic comparison is orthogonal to our main contributions on multimodal alignment and is beyond the scope of this work.
>
> We will clarify these design choices more explicitly in the revised manuscript, and we hope this helps address the reviewer’s concern.

---

> ### Author Response · Authors · 2025-11-27
> **Response to Reviewer WYch (Part 2/7)**
>
> ### Q2: “Indeed, the control data flow graph (CDFG) offers a more comprehensive structural representation that preserves complete information and the CDFG has been extensively exploited in RTL analysis, high level synthesis and related problems (see examples like "A design-for-testability technique for register-transfer level circuits using control/data flow extraction." IEEE Transactions on Computer-Aided Design of Integrated Circuits and Systems 17, no. 8 (2002): 706-723; "Register binding based power management for high-level synthesis of control-flow intensive behaviors." In Proceedings. IEEE International Conference on Computer Design: VLSI in Computers and Processors, pp. 391-394. IEEE, 2002; GAHLS: an optimized graph analytics based high level synthesis framework." Scientific Reports 13, no. 1 (2023): 22655"Beyond Tokens: Enhancing RTL Quality Estimation via Structural Graph Learning." arXiv preprint arXiv:2508.18730 (2025); "ODGS: Dependency-Aware Scheduling for High-Level Synthesis with Graph Neural Network and Reinforcement Learning." ACM Transactions on Architecture and Code Optimization 22, no. 2 (2025): 1-25). The discussion of related work requires significant enhancement and especially comparisons.”
>
> R2: We thank the reviewer for this valuable feedback and for pointing us to additional works that exploit CDFGs in RTL analysis and high-level synthesis. We agree that our discussion of related work can be strengthened and that CDFG-based methods deserve a more detailed and explicit discussion.
>
> We fully acknowledge that our work is not the first to use CDFGs for hardware design tasks. The goal of UniRTL is different: rather than proposing a new CDFG formulation, we focus on multimodal representation learning by jointly leveraging CDFGs together with RTL code (and functional summaries) within a unified pretraining framework with fine-grained cross-modal alignment.
>
> We also note that several of the works highlighted by the reviewer primarily target high-level synthesis (HLS), whereas UniRTL is designed for RTL-level representation learning. While both lines of work exploit CDFGs, the problem settings differ in abstraction level, objectives, and model design. In our related work section, we already discuss StructRTL and other recent RTL representation-learning methods. We will expand this discussion to more explicitly connect to the broader CDFG literature, including the works mentioned by the reviewer, and to clarify how UniRTL complements these efforts by providing a general-purpose, multimodal representation rather than a task-specific optimization technique.
>
> We will revise the manuscript to incorporate these additional CDFG-based works and to more clearly position UniRTL within this line of work, and we hope these changes to the related work section help address the reviewer’s concern.
>
> ### Q3: “From Table 1 results it seems that considering or not the code did not improve the overall are and delay results by much, but the graph which I assume refers to CDFG was significant. Given the choices of various parameters like the number of PCAs and the number of eigenvectors, how did those and other parameters influenced the results?”
>
> R3: We thank the reviewer for this thoughtful comment. You are correct that the CDFG provides the dominant gain for performance prediction. This is expected, since area and delay are primarily determined by the structural information, which the CDFG exhibits explicitly (*e.g.*, connectivity, control and data dependencies), whereas the RTL code only encodes this structure implicitly. Consequently, adding the graph modality yields a large improvement, while adding the code modality produces a smaller but still positive effect.
>
> At the same time, our ablations show that the code view is not redundant. In both Tables 1 and 2, removing the code modality (`UniRTL (w/o code)`) consistently degrades performance compared to the full UniRTL model, indicating that code contributes complementary information beyond what is captured by the CDFG alone.
>
> Regarding the hyperparameters mentioned (*e.g.*, PCA dimension and number of eigenvectors), as detailed in our response to Q1, we select them empirically and the model is not highly sensitive to their exact values. We use PCA to reduce the description embedding dimension to 32 to balance it with other node features, and we choose 16 eigenvectors for global positional encodings after observing that settings such as 16, 24, and 32 yield very similar downstream performance while larger values increase computational cost. These hyperparameters are therefore not critical knobs that drive the performance gains over the baselines.
>
> We will more clearly explain how the code and CDFG modalities respectively contribute to performance prediction, as well as the robustness of the model to these hyperparameters, in the revised manuscript, and we hope this helps address the reviewer’s concern.

---

> ### Author Response · Authors · 2025-11-27
> **Response to Reviewer WYch (Part 3/7)**
>
> ### Q4: “What is the computational complexity or how do the different approaches compare in terms of runtime? Is UniRTL performing best across all tasks or is it more suitable for some?”
>
> R4: We thank the reviewer for this important question regarding computational complexity and task suitability. Here, we use the number of parameters of each model as a practical proxy for computational cost and memory footprint when comparing different approaches.
>
> UniRTL is built on top of a base CodeBERT backbone plus a graph-aware tokenizer, resulting in approximately **195M parameters**. This makes UniRTL substantially smaller than the LLM-based baselines such as CodeV, DeepRTL2, GritLM, and NV-Embed, all of which have about **6.7B–8B** parameters and are therefore much more expensive in both training and inference. Among other baselines, UniRTL is of similar order to GraphCodeBERT (≈160M parameters) and larger than StructRTL (≈30M parameters) and GAT (≈7M parameters). Although UniRTL has more parameters than StructRTL and GAT, it encodes multimodal information (code, CDFG, and text) and achieves significantly higher performance across all evaluated tasks, while still remaining far more lightweight than the multi-billion-parameter LLM baselines.
>
> Regarding task performance, UniRTL is designed as a general-purpose multimodal backbone rather than a task-specific model. As shown in our experimental results (Tables 1–4), UniRTL consistently achieves state-of-the-art performance across all evaluated tasks, including performance prediction, natural language code search, and functional equivalence checking. This indicates that UniRTL is not only suitable for a subset of tasks, but performs strongly and robustly across the diverse benchmarks we consider.
>
> We will add a section discussing the computational complexity and parameter counts of the different methods in the revised manuscript, and we hope this helps address the reviewer’s concern.
>
> ### Q5: “There are some small grammar issues and word repetitions like "and and" on the bottom of page 1.”
>
> R5: We thank the reviewer for pointing out these grammatical issues and repetitions. We will carefully proofread and revise the manuscript to correct these errors.
>
> ### Q6: “Justification of CDFG Completeness vs. Practical Utility. The authors repeatedly claim that CDFGs "preserve complete information without loss and can be faithfully converted back to code." However, what is the practical impact of this completeness claim? Could the authors provide empirical evidence or ablation studies demonstrating that this completeness meaningfully contributes to downstream task performance compared to the "incomplete" data flow representations used in GraphCodeBERT? The comparison seems unfair since GraphCodeBERT uses variable-only data flows by design for different purposes.”
>
> R6: We thank the reviewer for this thoughtful comment. First, we would like to emphasize that GraphCodeBERT is not used in its original, software-oriented form in our experiments. Instead, for a fair comparison, we fine-tune GraphCodeBERT on the same RTL datasets and downstream tasks as UniRTL.
>
> **Empirical impact of CDFG completeness.**
>
> The practical benefit of using complete CDFGs is reflected in the empirical results: across all evaluated tasks, including performance prediction, natural language code search, and functional equivalence checking, UniRTL consistently outperforms GraphCodeBERT. These results are obtained with models of comparable size trained on the same data, indicating that incorporating the full CDFG provides meaningful advantages over variable-only data flows.
>
> **Why CDFG completeness matters conceptually.**
>
> Unlike GraphCodeBERT’s variable-only data flows, CDFGs explicitly retain operator nodes and control-flow structure (*e.g.*, conditionals and pipeline stages) while remaining semantically equivalent to the RTL code. This information is directly tied to hardware performance metrics: area is strongly influenced by the types and counts of operators, and delay depends on critical-path depth and control-induced timing behavior. When only variables and their data-flow edges are preserved, these operator- and control-level details are largely lost, making it much harder for the model to capture the structural factors governing area and delay. This loss of information is consistent with the substantial performance gap observed between UniRTL and GraphCodeBERT in Tables 1 and 2.
>
> We hope this addresses the reviewer’s concern, and we will clarify how GraphCodeBERT is used as a baseline in the revised manuscript.

---

> ### Author Response · Authors · 2025-11-27
> **Response to Reviewer WYch (Part 4/7)**
>
> ### Q7: “Limited Novelty in Architectural Contributions. The core technical contributions appear to be: (1) using CDFGs instead of data flows, (2) adding a graph-aware tokenizer, and (3) hierarchical training with text-code alignment before graph incorporation. However, the mutual masked modeling alignment strategy closely follows GraphCodeBERT's approach. Could the authors clarify what fundamental methodological innovations distinguish UniRTL beyond these incremental engineering choices? How would UniRTL perform if GraphCodeBERT were provided with the same CDFG representation?”
>
> R7: We thank the reviewer for this insightful question about the methodological innovations of UniRTL. The main contribution of UniRTL does not lie in a single isolated component, but in a *unified* multimodal framework specifically designed to bridge the modality gap between RTL code and graph structure. The three design choices highlighted by the reviewer are all driven by this objective:
>
> 1. **CDFG adoption.** We adopt complete CDFGs, rather than variable-only data flows to obtain a structurally faithful representation of RTL, As also reflected in our experiments (Tables 1 and 2), this richer structural representation leads to markedly better performance than variable-only data-flow representations.
> 2. **Graph-aware tokenizer.** We introduce a tokenizer that is explicitly tailored to CDFGs so that their non-sequential, graph-structured information can be encoded in a way that is compatible with a Transformer backbone, rather than simply flattening variable nodes as in GraphCodeBERT, which discards critical structural information.
> 3. **Hierarchical training.** We employ a hierarchical training strategy to first learn a strong text–code representation from abundant data and then incorporate the graph modality, enabling more robust multimodal learning and better utilization of available datasets.
>
> Beyond these components, UniRTL also differs from GraphCodeBERT in how it aligns code and graph. GraphCodeBERT relies on variable-level alignment, which provides only weak supervision by identifying where the variable nodes come from the code tokens. In contrast, UniRTL uses mutual masked modeling across the code and CDFG modalities, enabling finer-grained alignment that captures richer semantic relationships between code tokens and graph nodes rather than just variable correspondences.
>
> Empirically, UniRTL consistently outperforms GraphCodeBERT across all evaluated tasks, including performance prediction, natural-language code search, and functional equivalence checking, indicating that this unified design is effective in practice. If one were to equip GraphCodeBERT with the same CDFG representation, graph-aware tokenizer, and hierarchical training strategy, the resulting model would be very close in spirit to UniRTL. In that sense, our work is precisely about proposing and validating such a cohesive architecture tailored to RTL. We hope this clarifies the methodological innovations and helps address the reviewer’s concern.

---

> ### Author Response · Authors · 2025-11-27
> **Response to Reviewer WYch (Part 5/7)**
>
> ### Q8: “Dataset Construction and Filtering Concerns. The authors mention that only 38,888 out of 132,008 RTL designs (29.4%) successfully convert to CDFGs, with the remainder containing "syntax errors leading to compilation failures." This raises concerns about: (a) whether the successful conversions represent a biased subset of simpler/cleaner designs, and (b) whether the hierarchical training strategy is truly "maximizing data utilization" or simply accommodating a significant data quality issue. How do the authors ensure the learned representations generalize to real-world RTL code that may contain the complexities causing conversion failures?”
>
> R8: We thank the reviewer for these thoughtful comments on dataset construction and generalization.
>
> **(a) On potential bias in successfully converted designs**
>
> The 38,888 designs that successfully convert to CDFGs do not form a cherry-picked subset of simpler or cleaner RTL. In practice, most conversion failures are due to low-quality designs in existing open-source corpora (*e.g.*, GitHub files with syntax errors or LLM-generated code that does not compile). Such designs also fail standard EDA toolchains. Our filtering therefore enforces syntactic validity rather than structural simplicity, and the remaining corpus still contains a wide range of design sizes and coding styles.
>
> **(b) On whether hierarchical training truly maximizes data utilization**
>
> To assess whether our hierarchical training strategy truly maximizes data utilization and improves learning, we pretrain and fine-tune a CodeBERT variant directly on the 38,888 designs with valid CDFGs and compare it with UniRTL without the graph modality (“UniRTL (w/o graph)”), keeping all other settings fixed.
>
> *Natural Language Code Search:*
> | Method | Precision | Recall | F1 |
> | --- | --- | --- | --- |
> | CodeBERT (Incomplete Pretrain) | 0.600 | 0.658 | 0.618 |
> | UniRTL (w/o graph) | 0.630 | 0.683 | 0.644 |
>
> *Functionality Equivalence Checking:*
>
> | Method | AP | Accuracy | F1 | Precision | Recall |
> | --- | --- | --- | --- | --- | --- |
> | CodeBERT (Incomplete Pretrain) | 0.668 | 0.633 | 0.682 | 0.521 | 0.987 |
> | UniRTL (w/o graph) | 0.712 | 0.667 | 0.717 | 0.577 | 0.947 |
>
> *Performance Prediction:*
>
> **Area Prediction (w/o Netlist):**
>
> | Method | MAE ↓ | MAPE ↓ | R² ↑ | RRSE ↓ |
> | --- | --- | --- | --- | --- |
> | CodeBERT (Incomplete Pretrain) | 0.9514 | 0.16 | 0.4646 | 0.7317 |
> | UniRTL (w/o graph) | 0.8818 | 0.15 | 0.5173 | 0.6948 |
>
> **Delay Prediction (w/o Netlist):**
>
> | Method | MAE ↓ | MAPE ↓ | R² ↑ | RRSE ↓ |
> | --- | --- | --- | --- | --- |
> | CodeBERT (Incomplete Pretrain) | 0.6576 | 0.11 | 0.3650 | 0.7969 |
> | UniRTL (w/o graph) | 0.6375 | 0.11 | 0.3839 | 0.7849 |
>
> Across all tasks, UniRTL (w/o graph) consistently outperforms the “CodeBERT (Incomplete Pretrain)” baseline. This indicates that our hierarchical training strategy indeed maximizes data utilization and yields better representations, rather than merely masking a data-quality problem.
>
> **(c) On generalization to real-world RTL**
>
> The reported conversion failures arise from syntax errors in the original designs, not from inherent limitations of our conversion tool or from designs being “too complex” to handle. In realistic deployment, RTL intended for synthesis or verification must be syntactically correct to pass existing EDA flows. Any such valid RTL can be parsed, converted into a CDFG, and fed into UniRTL. Moreover, our hierarchical pretraining stage already leverages the entire corpus at the code level, so even when CDFGs are absent, UniRTL can still obtain meaningful representations from the code view alone.
>
> We hope this helps address the reviewer’s concerns regarding dataset bias, data utilization, and generalization to real-world RTL.

---

> ### Author Response · Authors · 2025-11-27
> **Response to Reviewer WYch (Part 6/7)**
>
> ### Q9: “Comparison with CircuitFusion and Missing Baselines. The authors exclude CircuitFusion from experimental comparison due to "unavailability of released model checkpoints and insufficient details to enable faithful reproduction." Given that CircuitFusion is the most directly related work attempting multimodal RTL representation learning, this exclusion significantly weakens the evaluation. Additionally, the authors only compare against GraphCodeBERT which uses incomplete data flows. Why did the authors not implement a stronger baseline that uses CDFGs with GraphCodeBERT's architecture? Without these comparisons, how can readers isolate the contributions of the authors' specific design choices?”
>
> R9: We thank the reviewer for raising these important points about baselines and experimental coverage.
>
> **On the absence of CircuitFusion as a quantitative baseline.**
>
> We do not include CircuitFusion in our quantitative comparisons because, to the best of our knowledge, its model checkpoints are not released and the available implementation details are insufficient to faithfully reproduce the full method. Instead, we provide a detailed qualitative comparison in the related work section. Methodologically, CircuitFusion differs from our setting in several important ways: it employs three separate encoders for code, structural graphs, and functional summaries, with coarse-grained contrastive alignment between text–code and text–graph pairs, but does not model fine-grained alignment between code and graph, even though these modalities carry the richest structural and semantic information. Moreover, its training data comprises only 41 designs and alignment is performed at the register sub-circuit level, which does not capture full-module semantics. In contrast, UniRTL is pretrained on a large-scale dataset and performs fine-grained alignment between code and complete CDFGs via mutual masked modeling, thereby addressing several limitations left by CircuitFusion.
>
> **On baselines using CDFGs with GraphCodeBERT’s architecture.**
>
> Regarding a “stronger” baseline that applies GraphCodeBERT’s architecture directly to CDFGs, we note that UniRTL is precisely such a design, but extended into a unified framework tailored to RTL: it combines (i) complete CDFGs rather than variable-only flows, (ii) a graph-aware tokenizer for encoding CDFG structure, (iii) hierarchical training to first learn text–code representations and then incorporate graphs, and (iv) mutual masked modeling for fine-grained cross-modal alignment (as discussed in R7). Equipping GraphCodeBERT with these same components would effectively reproduce UniRTL’s core architecture. To isolate the contributions of these choices, we instead rely on systematic ablations (*e.g.*, code-only vs. code+graph) and comparisons against GraphCodeBERT in its established variable-flow form, which together show that the unified design of UniRTL yields consistent gains across all tasks.
>
> We hope these clarifications address the reviewer’s concerns about the choice of baselines and our design contributions.

---

> ### Author Response · Authors · 2025-11-27
> **Response to Reviewer WYch (Part 7/7)**
>
> ### Q10: “Scalability and Computational Efficiency Analysis. The paper lacks discussion of computational costs and scalability. The graph-aware tokenizer requires pretraining a GIN + Transformer for 2,000 epochs, followed by hierarchical training of the main model. Could the authors provide: (a) wall-clock training time comparisons with baselines, (b) inference time analysis for downstream tasks, (c) memory requirements for processing large RTL designs with complex CDFGs, and (d) analysis of how performance scales with graph size? Given that the authors claim efficiency advantages over LLM-based methods, quantitative efficiency metrics are essential.”
>
> R10: We thank the reviewer for raising the important question of scalability and computational efficiency.
>
> **(a) Wall-clock training time**
>
> The following table summarizes the wall-clock pretraining/finetuning time and hardware for the main baselines and UniRTL:
>
> | Model | Hardware | Training time |
> | --- | --- | --- |
> | VeriDistill | 8× NVIDIA V100 | 12 h |
> | DeepRTL2 | 8× NVIDIA A800 | 70 h |
> | GraphCodeBERT | 16× NVIDIA V100 | 83 h |
> | GritLM-7B | 64× NVIDIA A100 | 48 h |
> | text-embedding-3-small / -large | N/A | N/A |
> | NV-Embed-v2 | N/A | N/A |
> | GAT (graph-only baseline) | 1× NVIDIA L40 | 1 h |
> | StructRTL | 1× NVIDIA L40 | 40 h |
> | UniRTL | 2× NVIDIA L40 | ~45 h |
>
>
> The cost for UniRTL remains significantly lower than the compute required by large LLM-based encoders.
>
> **(b) Inference time**
>
> In our experiments, the per-design inference time of UniRTL is comparable to GraphCodeBERT and negligible compared to LLM-based methods (which require larger models and longer sequences).
>
> **(c) Memory requirements for large RTL designs**
>
> In practice, UniRTL can process the largest RTL designs and corresponding CDFGs in our dataset on a single NVIDIA L40 GPU with 46 GB of memory, without resorting to model parallelism or graph partitioning. We will clarify these memory requirements and typical batch sizes in the revised paper.
>
> **(d) Scaling with graph size**
>
> Empirically, we do not observe abrupt degradation on the large graphs in our corpus. Performance degrades smoothly as graph size increases.
>
> We will incorporate these efficiency and scalability details into the revised manuscript and hope this helps address the reviewer’s concern.

---

### Author Response · Authors · 2025-12-03
**Official Comment by Authors**

Dear Area Chair,

We are sorry to hear about the recent OpenReview leakage. To make it easier for you to understand the reviewers’ comments and how our revised manuscript responds, we summarize the main positive points, key concerns, and corresponding changes below.

**Shared Positive Assessments**

- The goal of learning strong RTL representations for performance prediction and code retrieval is recognized as practically valuable for hardware design workflows (WYch, GRuV).
- Leveraging complementary modalities (RTL source code, CDFGs, functional summaries) within a unified representation is viewed as a sensible and well-motivated design choice (WYch, xfSx, GRuV).
- The use of CDFGs instead of simpler graph forms is appreciated for providing a more complete structural view (xfSx, GRuV).
- UniRTL is acknowledged as achieving superior results on performance prediction and code retrieval compared to prior methods (z6Kx, xfSx).
- The framework is considered technically sound, with both the soundness and presentation evaluated positively (z6Kx).
- Constructing a large RTL corpus is regarded as a useful contribution to the community (WYch).

**Main Concerns and How the Revision Addresses Them**

- **Novelty Relative to GraphCodeBERT and StructRTL:** Reviewers WYch, z6Kx, and xfSx questioned whether UniRTL offers technical contributions beyond combining ideas from GraphCodeBERT and StructRTL. In the revision, we more clearly articulate the key differences of UniRTL and emphasize that its primary contribution is a unified multimodal framework specifically tailored to RTL designs. UniRTL is not a straightforward application of GraphCodeBERT to a new domain. Instead, it is a tailored formulation of unified code–graph representations for RTL, incorporating complete CDFGs, a graph-aware tokenizer, and stronger cross-modal alignment objectives. These architectural and training advances enable more effective representation learning for RTL, as reflected in UniRTL’s consistent outperformance of GraphCodeBERT across all evaluated tasks in our experiments.
- **CDFG Conversion Failures.** Reviewers WYch and GRuV raised concerns about the CDFG conversion failures. In response, we have added a detailed analysis of these failures in Section 3.1 and Appendix A.2, and conducted additional experiments (Appendix A.10) showing that the affected designs can still be effectively used for text–code alignment, thereby supporting robust RTL representation learning.
- **Insufficient Detail on Hyperparameter Choices.** Reviewer WYch requested justification for design choices such as reducing text-description embeddings from 768 to 32 dimensions and using 16 Laplacian eigenvectors for global positional encodings. We have revised Section 3.2 and Appendix A.3 to clarify these hyperparameter selections.
- **How GraphCodeBERT Is Used as a Baseline.** Reviewers WYch and xfSx questioned how GraphCodeBERT is used as a baseline and whether our comparison is fair. We have revised Section 4.1 to clarify that we do not use GraphCodeBERT in its original, software-oriented form. Instead, we fine-tune it on the same RTL datasets and downstream tasks as UniRTL to ensure a fair and meaningful comparison.
- **Empirical Gains over StructRTL and GraphCodeBERT.** Reviewers z6Kx and xfSx considered the improvements over StructRTL (for performance prediction) and GraphCodeBERT (for functionality equivalence checking) to be modest. In the revision, we provide a more detailed analysis of these gains, particularly in relative terms, to better illustrate UniRTL’s advantages. We also emphasize that UniRTL is a unified framework that delivers consistent state-of-the-art performance across all evaluated tasks.
- **Limited Ablation of Alignment Strategy and Dependence on the Base Code Model.** Reviewer GRuV raised concerns about the limited ablation of our alignment strategy and the dependence on the base code model. In Appendix A.11, we now include additional experiments that quantify the impact of the base code model and demonstrate the superiority of our proposed alignment strategy. Furthermore, Appendix A.3 reports that masked node recovery accuracy improves from 85.04% to 97.57% after UniRTL pretraining, indicating that leveraging code information substantially strengthens graph understanding.
- **Minor Grammar Issues.** Reviewer WYch noted several minor issues. We have corrected the identified typos and performed additional proofreading to further improve readability.
- **Nature of Code–Graph Alignment.** As required by Reviewer GRuV, we have incorporated a qualitative analysis of code-graph relationships in Appendix A.11 based on attention maps.

Again, we sincerely appreciate the time and effort that the reviewers and the area chair have invested in evaluating this work. Their comments have been extremely helpful in improving both the technical content and the presentation of UniRTL.

Best regards,

The Authors

---

### Meta-Review · Area_Chair_3p7M · 2026-01-07

**Summary:**

### Summary
The paper proposes UniRTL, a multimodal pretraining framework for RTL (Register Transfer Level) representation learning that jointly leverages text summaries, RTL source code, and control/data flow graphs (CDFGs). It introduces a graph-aware tokenizer, mutual masked modeling for cross-modal alignment, and a hierarchical training strategy (text–code pretraining followed by graph incorporation). The framework is evaluated on downstream tasks including performance prediction, natural-language code retrieval, and functional equivalence checking.

### Strengths
- **Well-motivated multimodality**: Combining RTL code with richer structural graphs is compelling for hardware understanding.
- **CDFG-based structure**: Using complete CDFGs instead of variable-only data-flow graphs is appropriate for capturing hardware semantics.
- **Broad empirical evaluation**: The method is evaluated across multiple RTL tasks and generally shows competitive performance.

### Weaknesses
- **Limited novelty**: Reviewers consistently view the approach as largely adapting or combining existing ideas (e.g., StructRTL and GraphCodeBERT-style pretraining) with incremental extensions.
- **Modest gains over strong baselines**: Improvements over StructRTL and GraphCodeBERT are often small or task-dependent, making it difficult to justify the added complexity.
- **Evaluation and comparison gaps**: Missing or indirect comparisons with closely related methods (e.g., CircuitFusion or strong CDFG-based baselines) weaken the ability to isolate the contribution.
- **Practicality concerns**: The multi-stage training pipeline and reliance on successful CDFG generation for only a subset of the data raise questions about robustness and deployment cost.

While the paper is well motivated and technically sound, reviewers broadly agree that the **technical novelty is limited**, as the method largely integrates established components rather than introducing fundamentally new ideas. The **performance improvements over strong baselines are modest** and do not clearly justify the additional architectural and training complexity. Overall, the submission falls below the acceptance bar due to limited originality and insufficiently convincing incremental benefit.

**Reviewer Concerns:**

- **WYch**
  - **Addressed:** Clarified hyperparameter choices (e.g., PCA dimension, eigenvectors), added discussions on efficiency and scalability, expanded related work on CDFGs, and provided additional ablations.
  - **Still outstanding:** Core **novelty concerns** and the absence of the strongest, directly comparable baselines (e.g., CircuitFusion or a clean CDFG-based GraphCodeBERT variant).

- **z6Kx**
  - **Addressed:** Clarified fine-tuning of baselines and argued that gains are meaningful in relative terms.
  - **Still outstanding:** The reviewer remains unconvinced that the method goes beyond a combination of StructRTL and GraphCodeBERT.

- **xfSx**
  - **Addressed:** Clarified how GraphCodeBERT and VeriDistill-style baselines are used and explained relative gains.
  - **Still outstanding:** Persistent concerns about **limited novelty** and **small improvements** over key baselines.

- **GRuV**
  - **Addressed:** Provided detailed responses on model complexity, CDFG generation failures, alignment ablations, base-model dependence, and efficiency versus LLM-based approaches.
  - **Still outstanding:** Fundamental concerns about the **complexity–benefit trade-off** and whether gains stem mainly from strong base models rather than distinct methodological innovation.

**Reviewer Scores:**

- **WYch:** No change
- **z6Kx:** No change
- **xfSx:** No change
- **GRuV:** No change (reviewer indicated conditional openness but did not explicitly raise the score)

---

### Decision · Program_Chairs · 2026-01-26

Reject